# Characterization of a methyltransferase for iterative *N*-methylation at the leucinostatin termini in *Purpureocillium lilacinum*

Zixin Li[1,2,5], Yang Jiao[1,5], Jian Ling[1], Jianlong Zhao[1], Yuhong Yang[1], Zhenchuan Mao[1], Kaixiang Zhou[3], Wenzhao Wang[4], Bingyan Xie [1] ✉ & Yan Li [1] ✉

*N*-methyltransferase (NMT)-catalyzed methylation at the termini of nonribosomal peptides (NRPs) has rarely been reported. Here, we discover a fungal NMT LcsG for the iterative terminal *N*-methylation of a family of NRPs, leucinostatins. Gene deletion results suggest that LcsG is essential for leucinostatins methylation. Results from in vitro assays and HRESI-MS-MS analysis reveal the methylation sites as $NH_2$, $NHCH_3$ and $N(CH_3)_2$ in the C-terminus of various leucinostatins. LcsG catalysis yields new lipopeptides, some of which demonstrate effective antibiotic properties against the human pathogen *Cryptococcus neoformans* and the plant pathogen *Phytophthora infestans*. Multiple sequence alignments and site-directed mutagenesis of LcsG indicate the presence of a highly conserved SAM-binding pocket, along with two possible active site residues (D368 and D395). Molecular dynamics simulations show that the targeted N can dock between these two residues. Thus, this study suggests a method for increasing the variety of natural bioactivity of NPRs and a possible catalytic mechanism underlying the *N*-methylation of NRPs.

Nonribosomal peptides (NRPs), such as the antifungal agent echinocandins[1], the immunosuppressant cyclosporines[2,3], and the insecticide bassianolide[4], are a class of natural products with considerable biological and pharmaceutical potential. One common structural modification in these molecules is *N*-methylation. *N*-methylations contribute substantially to the properties of NRPs by modifying their structures[5] and influencing their bioactivities[6–8]. For example, the anthelmintic activities of cyclooctadepsipeptides with increased lipophilicity result from the introduction of homologous *N*-alkyl groups, which depend strongly on their nature of the *N*-methyl amino acid[9]. *N*-methylation of cyclosporin A may help increase oral availability[10] and stabilize the main conformation[7].

*N*-methyltransferases (NMTs) typically function as a domain within nonribosomal peptide synthetases (NRPSs)[11–14], and their catalytic sites are usually located on the main chain peptide bonds or active side chains of NRPs. A few NMTs are known as freestanding enzymes that catalyze the methylation of NRPs. For instance, in the production of bioactive

pentapeptides known as cycloaspeptides, the NRPS lacks NMT domains. Instead, an independent NMT is partnered with the NRPS to supply methylated substrates, preferentially incorporating methylated amino acids at two specific positions within the cycloaspeptides. However, there are few reports on the terminal *N*-methylation of NRPs[15]. The literature reviewed primarily focuses on single-site *N*-methylation within NRP structures. One study explores the *N*-methylation of the pipecolic acid moiety, which serves as the initial substrate for tubulysin synthesis. This methylation process is catalyzed by the NMT domain within the A-domain of its respective NRPS[16]. To our knowledge, the termini of NRPs have never before been shown to undergo a discrete NMT-catalyzed iterative *N*-methylation.

Leucinostatins are a family of lipopeptide antibiotics, derived from *Purpureocillium lilacinum*[17]. They exhibit a wide range of biological activities that affect multiple pathogens[18]. Furthermore, leucinostatins have been studied as potential anticancer agents and potent antiprotozoal agents[19–21], and their bioactivities inhibit mitochondrial function[22,23]. At least 24

[1]State Key Laboratory of Vegetable Biobreeding, Institute of Vegetables and Flowers, Chinese Academy of Agricultural Sciences, 100081 Beijing, China. [2]Microbial Processes and Interactions (MiPI), TERRA Teaching and Research Centre, Gembloux Agro-Bio Tech, University of Liège, 5030 Gembloux, Belgium. [3]Center for Advanced Materials Research, Advanced Institute of Natural Sciences, Beijing Normal University at Zhuhai, Zhuhai 519087, China. [4]State Key Laboratory of Mycology, Institute of Microbiology, Chinese Academy of Sciences, 100101 Beijing, China. [5]These authors contributed equally: Zixin Li, Yang Jiao. ✉e-mail: xiebingyan@caas.cn; liyan05@caas.cn

**Fig. 1 | Identification of the predicted methyltransferase LcsG. a** Structures of leucinostatins (**1–8**). MeHA methylhex-2-enoic acid, MePro 4-methyl-proline, AHMOD 2-amino-6-hydroxy-4-methyl-8-oxodecanoic acid, HyLeu hydroxyleucine, Aib aminoisobutyric acid, Leu leucine, βAla β-alanine. **b** Genetic organization of the leucinostatin BGC in *P. lilacinum* PLBJ-1. MT methyltransferase, PKS polyketide synthase, NRPS nonribosomal peptide synthetase. **c** LC–MS analysis of the lcsG knockout (*ΔlcsG*) mutant and the wild-type (WT) strain.

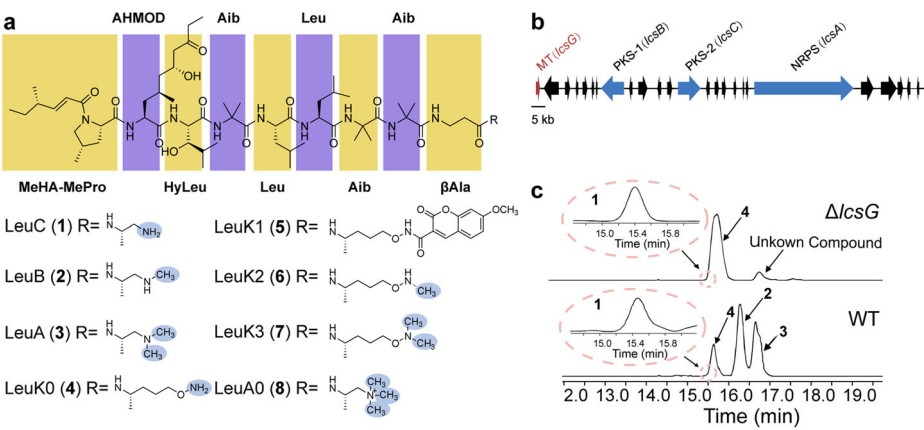

leucinostatin homologs have been isolated and characterized[24]. Their backbone chains are assembled by an NRPS with nine amino acids and an atypical amino moiety at the C-terminus, linked by peptide linkages[25]. The C-termini of leucinostatins are methylated to varying degrees. Specifically, the C-terminus of leucinostatin C (**1**, LeuC) is propane-1,2-diamine (PD), while those of leucinostatin B (**2**, LeuB) and leucinostatin A (**3**, LeuA) are protected by *N*-methylpropane-1,2-diamine (MPD), and $N_1$, $N_1$-dimethylpropane-1,2-diamine (DMPD), respectively (Fig. 1a). However, the biosynthetic mechanisms involved in the generation of diverse C-termini remain unknown.

In this work, we identify a discrete NMT, LcsG, from the biological control fungus *Purpureocillium lilacinum* PLBJ-1 that catalyzes the incorporation of a unique moiety located at the terminus of an NRP[26]. Deleting the *lcsG* gene led to the disappearance of leucinostatins with methylated terminus. Subsequent in vitro enzyme activity assays and structural elucidation of the products demonstrated that LcsG is involved in the iterative methylation of terminal-free amines in leucinostatins. Moreover, structure–function relationship analysis provided a probable insight into the catalytic mechanism of LcsG. Additionally, we obtained new leucinostatins from the LcsG-catalyzed reaction, that can inhibit the growth of the human pathogen *Cryptococcus neoformans* and the plant pathogen *Phytophthora infestans*.

## Results
### Identification of LcsG from *Purpureocillium lilacinum*
Our previous studies demonstrated that the *P. lilacinum* PLBJ-1 strain harbors the leucinostatin biosynthesis-related gene cluster (BGC)[26], and the *lcsG* gene, which was predicted to function as a methyltransferase attracted our interest (Fig. 1b). LcsG is equipped with the Methyltransf_2 domain, which is commonly found in *O*-methyltransferases (OMTs). The top hit in its pBLAST search shared 31.65% sequence identity with an OMT called VdtC (A0A443HJY8.1)[27]. However, despite this prediction, we did not find any *O*-methylated units in leucinostatins. This inconsistency prompted us to investigate the function of LcsG.

To determine the function of LcsG in leucinostatin biosynthesis, we constructed deletion mutants (*ΔlcsG*) and overexpression mutants (OE*lcsG*) of *P. lilacinum* PLBJ-1 (Supplementary Figs. 1–3). Following growth on a productive medium and production extraction, LC–MS analysis was performed, revealing obvious differences between the deletion mutant and wild-type strains (WT) (Fig. 1c); in contrast, there were no obvious differences between the overexpression mutants (OE1, OE2, which correspond to strains overexpressing the *lcsG* gene) and the WT (Supplementary Fig. 3). LC–MS analysis of the WT extracts revealed that the $m/z$ $[M + H]^+$ values of the four peaks (**1–4**) were 1190.8133, 1218.8439, 1204.8247, and 1234.8388, respectively (Supplementary Fig. 4). High-resolution electrospray ionization mass spectrometry (HRESI-MS-MS) analysis (Supplementary Table 1 and Supplementary Fig. 5) confirmed that peaks **1–3** were attributed to the LeuC, LeuB, and LeuA, respectively. Based on the HRESI-MS-MS results and the same molecular weight obtained, peak **4** was initially presumed to be

leucinostatin K (LeuK), a compound isolated from *Paecilomyces lilacinus* (synonym *Purpureocillium lilacinum*)[28]. However, the NMR analysis suggested that peak **4** was not the known compound LeuK but rather a compound derived from LeuC whose C-terminus was -NH-CH$_2$-CH$_2$-OH or -CH$_2$-CH$_2$-O-NH$_2$. As its structure was not fully consistent with that of LeuC, the compound has not been previously reported, we designated it as leucinostatin K0 (**4**, LeuK0) since the name "K0" was unclaimed (Supplementary Figs. 6–8). A specific *N*-hydroxysuccinimide (NHS)-ester reaction was employed to verify the free amine in LeuK0, and then the product compound LeuK1 (**5**) with NHS ester labeling was detected (Supplementary Fig. 9), which confirmed the structure of LueK0 (**4**) as shown in Fig. 1a. The deletion of *lcsG* led to the abolishment of LeuB and LeuA, both of which have methylated termini, suggesting that the deletion blocked the formation of the methylated C-terminal amines. Moreover, LeuK0 and an unidentified compound that shared the same C-terminus (Supplementary Table 1) persisted in the deletion mutant (Fig. 1c). Therefore, LcsG was inferred to play an essential role in the biosynthesis of terminal amines of leucinostatins.

### LcsG functions as a SAM-dependent methyltransferase
Sequence analysis suggested that LcsG is an *S*-adenosyl-L-methionine (SAM)-dependent methyltransferase. To clarify its biochemical function, we expressed *lcsG* in *E. coli* ArcticExpress (DE3) and purified the recombinant protein LcsG, which was tagged with His$_6$ at both the N and C termini, using nickel affinity chromatography (Fig. 2a). Obtaining pure LeuB and LeuC in the laboratory is exceptionally difficult. The enzymatic activity was assayed by using LeuK0 (**4**) and LeuA (**3**) as the substrates. The S-adenosyl-L-homocysteine (SAH) and three new product peaks, LeuK2 (**6**), LeuK3 (**7**) and LeuA0 (**8**), were detected only in the presence of the substrate, LcsG, and SAM (Supplementary Fig. 10 and Fig. 2b). In contrast, omitting LcsG or SAM resulted in no product formation. LC-MS analysis confirmed that the $[M + H]^+$ values of the **6–8** ions were 1248.8533, 1262.8711, and 1232.8595, respectively (Supplementary Fig. 11). These molecular weights indicated that LeuK2 (**6**) and LeuA0 (**8**) are the methylated products of LeuK0 (**4**) and LeuA (**3**), respectively, and that LeuK3 (**7**) is the dimethylated product of LeuK0 (**4**). These results indicated that LcsG is likely a SAM-dependent methyltransferase.

The kinetic activity of LcsG was analyzed to gain further insight into its methylation activity. We conducted several single-factor enzymatic assays. The ethyl acetate (EtOAc) extracts of the WT PLBJ-1 strain cultures, which inherently contained LeuB (**2**), LeuA (**3**), and LeuK0 (**4**), served as substrates. We also established a control group without the addition of LcsG and employed LC-MS analysis to quantify each component in the experimental and control groups. The variations in component levels between the experimental and control groups were compared to characterize the changes induced by LcsG catalysis. Positive values indicate that a component's level in the experimental group is greater than that in the control group, suggesting that a component accumulates during catalysis. Conversely, negative values suggest that the level of the component in the experimental group was lower

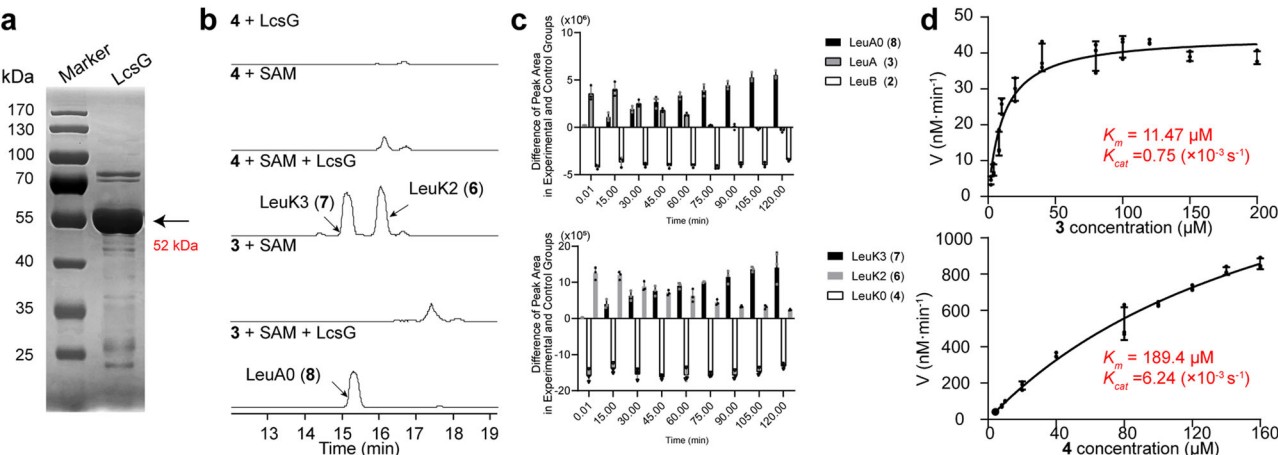

**Fig. 2 | Verification that LcsG is involved in the methylation of leucinostatins.** **a** SDS-PAGE analysis of the recombinant protein LcsG. **b** In vitro LcsG activity analysis using LeuK0 (**4**) and LeuA (**3**) as the substrates and the $[M + H]^+$ ions of three new product peaks at positions **6**–**8**. **c** Time dependency of the variations in

each component. **d** Kinetic analysis of LcsG using LeuA (**3**, top) and LeuK0 (**4**, bottom) as substrates. All the data are represented as the means of $n = 3$ biologically independent samples, and the error bars show the standard deviations (**c**, **d**).

**Fig. 3 | Structural elucidation of LeuA0 (8), LeuK2 (6), and LeuK3 (7).** **a** LC-MS analysis of the $N$-methylation of LeuA (**3**) to the same trimethyl-lammonium compound (**8**, LeuA0) by LcsG and $CH_3I$. **b** LC-MS analysis of two methylation reactions using LeuK0 (**4**) as the substrate and LcsG and $CH_3I$. **c** Structures and the HERSI-MS-MS data of LeuA0 (**8**), LeuK0 (**4**), LeuK2 (**6**), and LeuK3 (**7**). **d** Growth inhibition of eukaryotic microorganisms by 25 µg/well LeuK0 (**4**), LeuA0 (**8**), and LeuK3 (**7**) according to the agar diffusion assay.

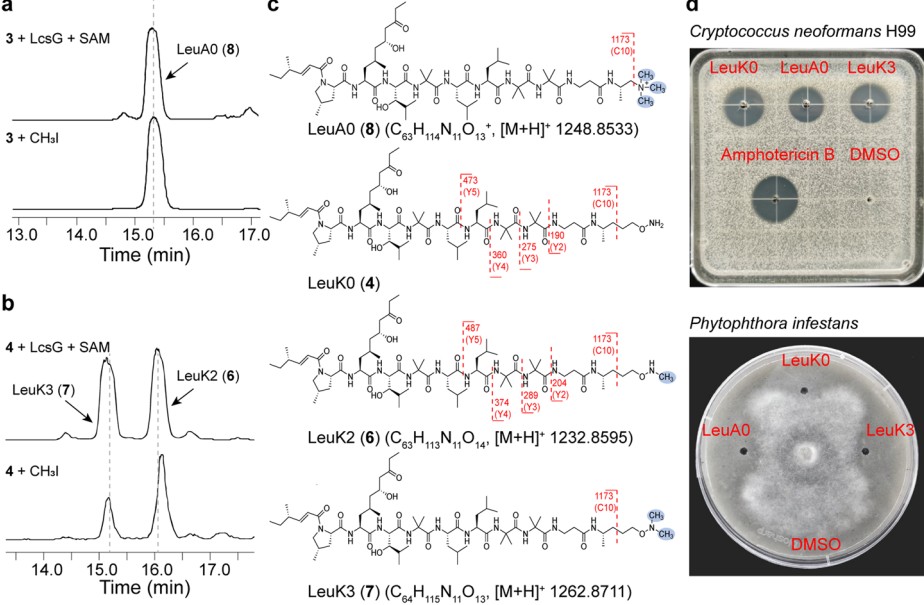

than that in the control group, indicating that a component is consumed during catalysis. The time dependency of the LcsG-catalyzed reaction showed that LeuA0 (**8**) and LeuK3 (**7**) accumulated nearly linearly with time and appeared almost immediately after the reaction started. LeuB (**2**) and LeuK0 (**4**) were nearly completely consumed after the beginning of the reaction, while the amount of LeuA (**3**) and LeuK2 (**6**) increased early on and decreased (Fig. 2c). These results suggested that LcsG is an iterative methyltransferase, and the reaction sequences could be LeuB-LeuA-LeuA0 and LeuK0-LeuK2-LeuK3. The optimal pH and temperature for producing the final products LeuA0 and LeuK3 were determined (Supplementary Fig. 12), followed by measurements of the initial rates at substrate concentrations ranging from 0–200 µM. The initial rate data were measured by LC-MS and fitted to the Michaelis−Menten equation to determine the kinetic parameters (Fig. 2d and Supplementary Fig. 13). The $K_{cat}/K_m$ values obtained for LeuA (**3**) and LeuK0 (**4**) were 65.39 $s^{-1} M^{-1}$ and 32.94 $s^{-1} M^{-1}$, respectively.

**Characterizations of the products of the LcsG-catalyzed reaction**
Next, we performed HRESI-MS-MS analysis to elucidate the structures of compounds LeuK2 (**6**), LeuK3 (**7**) and LeuA0 (**8**), we turned to HRESI-MS-

MS analysis. The $m/z$ values of the fragments of each leucinostatin are presented in Supplementary Table 1. Comparisons of the MS-MS data of LeuA0 (**8**) with those of LeuA (**3**) and LeuB (**2**) revealed that the spectrum of LeuA0 (**8**) showed remarkably similar fragments to those of LeuA (**3**) and LeuB (**2**) (Supplementary Fig. 14). Specifically, they shared one fragment with a $m/z$ value of 1173. This ion was deduced to be the $[M + H]^+$ ion of fragment C10, indicating the possible methylation site of the C-terminal amine. Therefore, LeuA0 (**8**) was concluded to be a trimethylammonium compound in which the terminal amine carried a positive charge and three methyl groups. This predicted structure is identical to a previously identified structure[29], which was obtained by treating LeuA (**3**) with methyl iodide. This reaction afforded the same product as the enzymatic reaction (Fig. 3a), confirming the structure of LeuA0 (Fig. 3c). Combined with its molecular weight, LeuA0 (**8**) was assigned the molecular formula $C_{63}H_{114}N_{11}O_{13}^+$.

Treating LeuK0 (**4**) with methyl iodide produced LeuK2 (**6**) and LeuK3 (**7**) as shown in Fig. 3b. Based on their HRESIMS data, LeuK2 (**6**) and LeuK3 (**7**) have molecular formulas of $C_{63}H_{113}N_{11}O_{14}$ and $C_{64}H_{115}N_{11}O_{14}$, respectively. MS-MS experiments revealed similar fragments to LeuK0, with differences in Y-type fragments. Notably, the Y-type fragments in LeuK2 (**6**)

**Fig. 4 | Catalytic sites of LcsG. a** SAH binding sites of OxaC (PDB code: 5w7p, marked in pink), LepI (PDB code: 6ix7, marked in blue), and LcsG (marked in golden). **b** Modified LeuA binding sites of LcsG (Maestro, Schrödinger, LLC). **c** Comparison of wild-type LcsG and mutated LcsG-mediated methylation. The yield of the product was quantified by comparison with the peak area of the product catalyzed by LcsG (100%). All the data are represented as the means of $n = 3$ biologically independent samples and the error bars show the standard deviations.

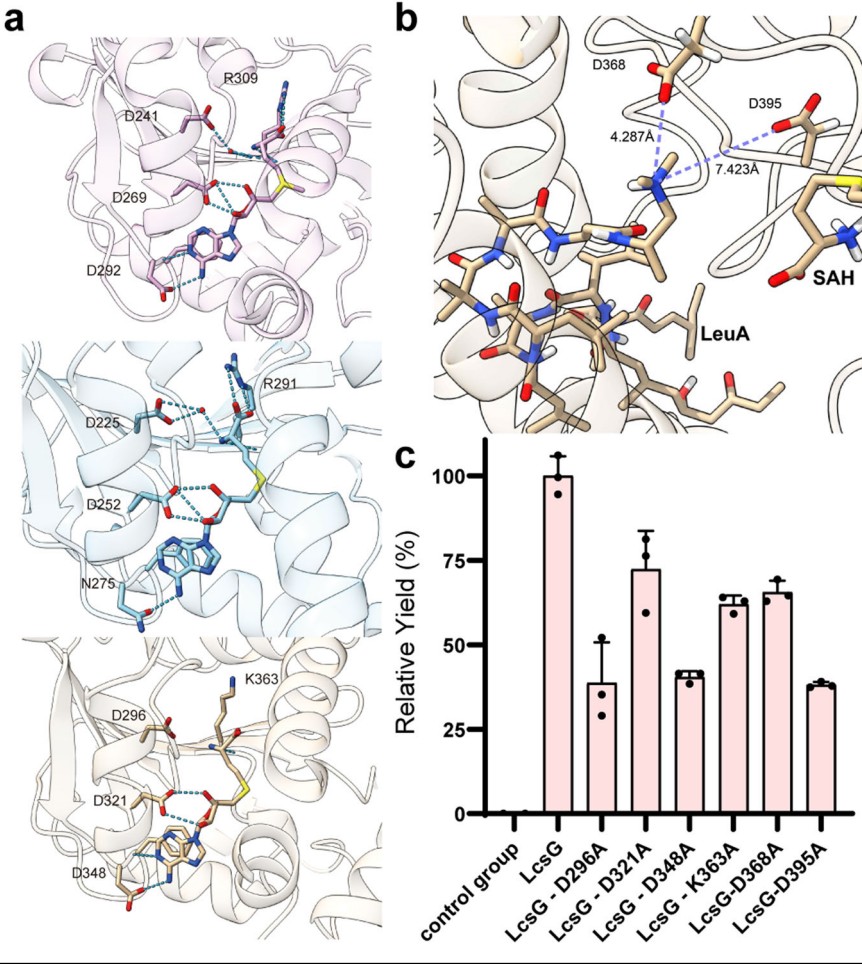

are +14 mass units larger than those in LeuK0 (**4**). For example, *m/z* values of 190 and 204 correspond to $[M + H]^+$ ions of Y2 fragments in LeuK0 and LeuK2 (**6**), respectively. Furthermore, LeuK2 (**6**) and LeuK3 (**7**) produced an ion at *m/z* 1173 (C10), indicating that the terminal amine of LeuK2 was methylated LeuK2 (**6**) (Supplementary Fig. 15). These findings revealed LeuK2 (**6**) and LeuK3 (**7**) as new leucinostatins, as depicted in Fig. 3c. Overall, these results confirm that LcsG, as an NMT, can iteratively catalyze the methylation of NRP terminal amines.

**Antimicrobial evaluation of leucinostatins**

Leucinostatins are well-known antibiotics. We purified these compounds by semipreparative HPLC and successfully generated LeuA0 (**8**), LeuK0 (**4**) and LeuK3 (**7**) in sufficient quantities for the antimicrobial assay. The inhibitory activity of these compounds against the drug-resistant strain *C. neoformans* H99 and the plant pathogen *P. infestans* was determined using agar diffusion assays. Notably, previous studies have demonstrated the inhibitory effects of LeuA and LeuB on both *C. neoformans*[18] and *P. infestans*[26]. All of these leucinostatins had inhibitory effects on these pathogens (Fig. 3d). Moreover, the anti-*C. neoformans* MICs value of the two methylated products, LeuA0 (**8**, 25.8 μg/ml) and LeuK3 (**7**, 25.8 μg/ml), were four and two times lower than those of their parent products, LeuA (**3**, 102.4 μg/ml) and LeuK0 (**4**, 51.2 μg/ml), respectively (Supplementary Fig. 16), which indicated that *N*-methylation at the terminus of leucinostatins can improve their antimicrobial efficiency.

**Catalytic mechanism of the *N*-methyltransferase LcsG**

We then identified the catalytic residues in LcsG. Local multiple sequence alignments revealed that LcsG shares a conserved SAH/SAM-binding motif (D296, D321, D348, and K363) and two possible catalytic residues (D368 and D395) (Supplementary Fig. 17a). Despite many attempts, we were

unable to obtain a crystal of the LcsG protein suitable for X-ray crystal-lographic analysis. As an alternative, we employed an artificial intelligence (AI) method, Uni-Fold, to approximate a model of LcsG. The overall structure of LcsG with color-coded pLDDT scores is presented in Supplementary Fig. 18a, and the six predicted residues are depicted in Supplementary Fig. 18b. Additionally, Supplementary Table 2 provides the pLDDT scores for these six predicted residues. These results indicated that AI predicts the protein structure with high confidence, including the active site residues that are proposed to be relevant for substrate binding and catalysis. Structural alignments between LcsG and its homologs are shown in Supplementary Fig. 17b, with corresponding RMSD values detailed in Supplementary Table 3. The overall structure of LcsG involved a typical Class I methyltransferase fold; the N-terminus appears to be responsible for dimerization and substrate binding and the C-terminus appears to be responsible for SAM-binding (Supplementary Fig. 18b). To determine the structure–function relationship of LcsG, we conducted a molecular docking analysis using the predicted LcsG structure and SAH. SAH was docked into the LcsG structure model binding pocket, and the hydrogen bond network that mediates SAH binding was present in the final docking position.

As shown in Fig. 4a, potential hydrogen bond interactions between SAH and residues D296, D321, D348, and D363 were detected, which was consistent with the results of multiple sequence alignment (Supplementary Fig. 18c). The accurate stereo-structure structures of nonpeptide leucinos-tatins are difficult to predict because they are composed of seven non-standard and unusual α-amino acid residues. Based on the crystal structures of the analogs helioferin A[21] and ZHAWOC6027[30] (Supplementary Fig. 19), a structural model of LeuA was predicted, and LeuA was docked into the LcsG structure via DiffDock[31]. In the first ranked result, the *N* atom that becomes methylated occupies the position between D368, D395 and SAH (Supplementary Fig. 20). Subsequent molecular dynamics (MD)

simulations were used to further explore the stabilities of our protein–ligand complex and the importance of catalytic site residues. After 500 ns of equilibration (Supplementary Fig. 21), 3 independent 500 ns MD simulations were performed (Supplementary Table 4 and Supplementary Fig. 22). The results showed that the positively charged N is located between D395 and D368, and the terminal N of LcsG could form a stable complex system with D368 (Fig. 4b). Nevertheless, the results provided insight into the importance of D368. In some snapshots, the ligand formed salt bridges with D368 and D395 (Supplementary Fig. 23).

To verify this result, we mutated D296, D321, D348, K363, D368, and D395 to Ala in LcsG (Supplementary Fig. 24). Biochemical assays of these mutants were then performed using LeuA as substrate. After 1 h of incubation, the conversion of LeuA (**3**) to LeuA0 (**8**) decreased for all the mutants (Fig. 4c). These findings are consistent with our earlier in vitro assays and docking and MD simulation results. These findings indicate that the SAM-binding site of LcsG is involved in leucinostatin methylation and provide additional support that LcsG is a SAM-dependent methyltransferase. Regarding the enzymatic mechanism of methyltransferases, numerous studies have proposed a reaction mechanism for OMT, involving a His/Glu dyad and an Asp residue[28]. The Glu residue was placed near the His residue, and the His residue was activated to deprotonate the hydroxyl group in the substrate. The Asp residue was shown to interact with the substrate to improve binding. The deprotonated hydroxyl group functions as a good nucleophile to attack SAM[32], which is a methyl donor, to form the *O*-methylation product. In the case of NMT, similar but different mechanisms have been proposed[33]. A QM/MM study on the catalytic mechanism of phenylethanolamine NMT[34] revealed that unlike OMT, a Glu residue was employed to deprotonate the protonated amine in the substrate to form a nucleophile. Then, the methyl group was transferred from the methyl-donor SAM to the deprotonated amine group. Unlike reactions catalyzed by OMT, a His/Glu dyad was not needed for NMT. The lone electron pair of the *N* atom on the dimethylamine group can undergo a nucleophilic attack. We proposed that the protonated dimethylamine group in leucinostatins might be coordinated and deprotonated by two negatively charged residues (D368 and D395). A nucleophilic attack between dimethylamine and SAM followed, and the methyl group was transferred from SAM to leucinostatins. Compared to wild-type LcsG, the D368A and D395A mutants exhibited markedly obvious decreased methylation (Fig. 4c), indicating that these two residues contribute notably to the substrate binding. For LcsG, the mutation data, docking results and MD simulation results collectively gave a hint that its reaction mechanism is similar to that of phenylethanolamine NMT.

## Discussion

NMTs are important for the biotransformation of bioactive molecules. *N*-methylation can modulate the activity of signaling molecules and participate in the biosynthesis of natural products. NMTs are of great interest because site-specific modifications are crucial for the bioactivity and biosynthesis of natural products. In this study, we successfully determined the in vitro activity of the NMT LcsG, which is involved in the iterative *N*-methylation of the unique terminal unit of leucinostatins. Few NMTs are known to iteratively transfer methyl groups to natural products (Supplementary Table 5). The NMTs EgtD[35], PfPMT[36], BANMT[37] and NRMT[38] were reported to catalyze the *N*-trimethylation of corresponding substrates progressively. Moreover, PsiM[39], RosA[40], RedM[41], OxyT[42], and AMNMT[43] were shown to catalyze the two methylation reactions. Although the NMTs involved in the biosynthesis of plantazolicin-class ribosomally synthesized and posttranslationally modified peptides (RiPPs) are responsible for dimethylation, the corresponding monomethylated products have never been detected[8,44].

In functional OMTs, the charge relay system involving the His-Glu catalytic dyad is highly conserved[45]. The His serves as the general base, facilitating the deprotonation of the substrate nucleophile, typically the hydroxyl group[46]. Taking the proteins analyzed in our previous multiple sequence alignments as examples (Supplementary Fig. 17a), the basic

residues in ChOMT[47], MmcR[48], and OxaC (H278, H259, H313) contrast with LcsG's Y (at position 367). Among the identified structural homologs of LcsG in the PDB database, RosA and RedB are the only two NMTs with reported crystal structures and active sites; RosA and RedB feature S253 and E255 as their corresponding residues, respectively. To expand the search, we selected fungal-derived methyltransferases similar to LcsG, annotated by the same Pfam (PF00891), for analysis (Supplementary Table 6). Sequence comparison of OMTs and NMTs mentioned in Supplementary Tables 5 and 6 revealed that most of these OMTs have basic residues, while NMTs exhibit neutral or acidic residues at the corresponding positions (Supplementary Fig. 25). As hypothesized in our study, in the LcsG-catalyzed *N*-methyl transfer reaction, which involves a nucleophilic attack by the terminal N lone electron pair of leucinostatin on the reactive sulfonium methyl group of SAM, the *N*-methyl transfer reaction may not require a basic residue for deprotonation.

To elucidate the relationship between LcsG and other methyltransferases, we constructed a phylogenetic tree that included LcsG and experimentally confirmed similar MT sequences (Supplementary Data 1) and NMTs listed in Supplementary Table 5 (Supplementary Fig. 26a). This tree is divided into four main phylogenetic groups representing genes from fungi, bacteria, animals, and plants. Notably, OMTs and NMTs did not form distinct evolutionary groups, with LcsG appearing on an independent branch. An exception was observed in *Streptomyces* sp., in which OMT (P16559) and NMT (RosA) showed a close phylogenetic relationship and high similarity in sequences (Supplementary Fig. 26b); this result indicated that a gene duplication event may have led to divergence of OMT and NMT after the speciation of *Streptomyces*. Consistent with previous findings, the residue corresponding to Y367 in LcsG is H in OMT and E in NMT. (Supplementary Fig. 26b). Further investigation revealed a sequence similarity network (SSN) for LcsG and sequences mentioned in Supplementary Table 6 (Supplementary Fig. 27a). The SSN findings were consistent with the phylogenetic tree, showing no clear separation between OMTs and NMTs. In the network, NMTs, including LcsG, exhibited more connections with OMTs. Specifically, NMT Fsa4 clustered with three other NMTs (PynC, EqxD, and Phm5), likely due to similarities in their substrates (Supplementary Fig. 27b). Apart from LcsG, we propose that some sequences annotated as OMTs in genomic databases may actually be NMTs. Differences in some specific acidic residues at key positions in OMTs and NMTs, such as Y versus H, can serve as an indicator. This bioinformatic analysis indicated a possible evolutionary link between OMTs and NMTs, and also emphasized the important functions of those specific amino acids in OMTs and NMTs.

NRPSs are well-known megaenzymes that consist of sequential domains. The peptide is elongated, followed by release from the terminal module, which includes condensation domains (C$_T$), reductase domains (R), Dieckmann cyclase domains (D), and thioesterase domains (TE)[49]. According to the analysis of the antiSMASH 2.0 and pBLAST results (Supplementary Table 7), the terminal module of the NRPS in the biosynthesis-related gene cluster (LcsA) should be an R domain that can release the peptide from the NRPS by hydrolysis; thus, the C-terminus of leucinostatins is normally an aldehyde group[50–53] or a hydroxyl group[54–56] (Supplementary Fig. 28). However, we have not observed these similar structures among the existing characterized leucinostatins. Based on these reports and the NMR results of LeuK0, we initially deduced the C-terminus of LeuK0 to be -NH-CH$_2$-CH$_2$-OH, but this hypothesis was rejected by subsequent NHS ester reactions and enzyme assays.

In conclusion, an OMT-like enzyme from *P. lilacinum*, LcsG, was identified as a discrete SAM-dependent NMT that can iteratively catalyze the formation of primary amines, secondary amines, and tertiary amines in the unique terminal unit of leucinostatins. Furthermore, one new secondary metabolite (LeuK0) and two enzymatic products (LeuK2 and LeuK3) were identified as new leucinostatins. In addition, the methylated compounds were observed to display greater antimicrobial activity than that of their parent molecules. To our knowledge, LcsG is a rare NMT that can methylate the terminal residues of NRPs. We expect that the results of this study will

provide deeper insights into the mechanisms underlying the *N*-methylation of peptides and increase the possibility of engineering new methylated molecules for exploring more potent antibiotics.

## Methods

### Strains and culture conditions

The strains used in this study are listed in Supplementary Table 8. *Purpureocillium lilacinum* strain PLBJ-1 (CGMCC3.17492)[27,57,58], was isolated from tomato roots in Beijing, and its transformants were cultured at 28 °C in potato dextrose agar (PDA) or potato dextrose broth (PDB) supplemented with 400 μg/ml geneticin (G418). *Escherichia coli* Trelief 5α (Tsingke, China) was cultured at 37 °C in Luria–Bertani (LB) broth supplemented with appropriate antibiotics. *E. coli* ArcticExpress (DE3) (Agilent Technologies) was used to express LcsG proteins. *E. coli* ArcticExpress (DE3) was cultivated at 37 °C in LB broth supplemented with appropriate antibiotics for growth, followed by growth at 11 °C to induce the recombinant protein.

### DNA and RNA isolation

The mycelia of PLBJ-1 and the mutants were harvested via filtration after cultivation in 2 ml of potato dextrose broth (PDB) at 28 °C for 24 h. The genomic DNA was extracted using a Qiagen DNeasy Kit. RNA was extracted using a TRIzol reagent (Takara, Japan) following the manufacturer's protocol, and the RNA was cultivated on PDA at 28 °C for 3 days.

### Gene cloning and plasmid construction

The oligonucleotide sequences for PCR primers are listed in Supplementary Table 9. PCRs were performed using 2 × Phanta Max Master Mix Polymerase (P525, Vazyme Biotech Co., Ltd, China) and Q5 High-Fidelity DNA Polymerase (New England Biolabs, USA). The plasmids used are listed in Supplementary Table 8. To construct the deletion cassette of *lcsG*, approximately 1 kb DNA fragments located upstream and downstream of the *lcsG* coding region were amplified from the gDNA of PLBJ-1 and named *lcsGup* and *lcsGdown*, respectively. Two fragments and the selection marker gene *neo* were integrated into the *Kpn*I/*Bam*HI-cleaved vector pKOV21 via the digestion-ligation method by using T4 DNA Ligase (Thermo Fisher Scientific, USA) to generate the deletion plasmid pKOV21-ko*lcsG*.

For the overexpression of *lcsG* in PLBJ-1, the *lcsG* gene was amplified from the cDNA of PLBJ-1. The selection marker gene *neo* and the terminator *TrpC* were amplified from the KSTNP vector. This fragment and two restriction enzyme cutting sites, *Pme*I and *Pac*I, were integrated into the pEASY vector by using the *pEASY®*-Blunt cloning Kit (TransGen Biotech, China) to generate the intermediate vector pEASY-*neoTrpC*. Then the strong promoter *gpdA* was amplified from the PCH-sGFP vector and integrated into the *Not*I/*Apa*I-cleaved vector pEASY-*neoTrpC* via the digestion-ligation method by using DNA T4 ligase to generate the vector pGNT. Afterward, the *lcsG* gene was integrated into the *Pme*I/*Not*I-cleaved vector pGNT by using *pEASY®*-Basic Seamless Cloning and Assembly Kit (TransGen Biotech, China) to create the overexpression vector pGNT-*lcsG*. The recombinant protein LcsG expression vector pACYC-*lcsG* was generated by integrating the *lcsG* gene from the PLBJ-1 cDNA into the protein expression vector pACYCDuet-1 by using the quick-change method[59]. The mutated LcsG protein vectors were obtained by using a QuickMutation™ Site-Directed Mutagenesis Kit (D02065, Beyotime Biotechnology, China) following the manufacturer's instructions. The following sites were used for mutagenesis of the pACYC-*lcsG* plasmid: D296A (AGT to GCT), D321A (GAT to GCT), D348A (GAC to GCC), K363A (CAA to CGC), D368A (GAC to GCC), and D395A (GAT to GCT). Mutations were confirmed by sequencing.

### PEG-mediated fungal transformation

The split-marker strategy was used to disrupt the *lcsG* gene. The DNA fragments *lcsGup-ne* (*lcsGup* and the first half of *neo*) and *eo-lcsGdown* (the second half of *neo* and *lcsGdown*) were amplified from pKOV21-ko*lcsG*. The two fragments of *neo* overlapped by 667 bp. For polyethylene glycol (PEG)-mediated fungal transformation, $10^8$ spores of PLBJ-1 was cultured

in 200 ml of PDB medium at 28 °C and 150 rpm for 18 h. The fungal germlings were harvested using a 4-layer Miracloth filter (Solarbio, China). After washing with 0.7 M NaCl, the mycelia were treated with 2 μg/ml driselase (Sigma) at 28 °C and 150 rpm for 4 h to obtain protoplasts. Microscopic examination was conducted to assess the protoplast status. Following filtration with the 4-layer Miracloth and thorough washing with STC buffer, the protoplasts were collected by centrifugation at 4 °C, 4000 rpm for 15 min, and resuspended to a concentration of $10^6$/ml in STC buffer. Five microgram of the two DNA fragments were transformed into 100 μl of PLBJ-1 protoplasts. G418-resistant colonies were selected after culture on PDA at 28 °C for 1 day. The candidate transformants were picked and inoculated onto new PDA plates supplemented with 400 μg/ml G418 (Inalco, USA) for 3–5 days. These transformants were verified via diagnostic PCR with primers. For the overexpression of *lcsG*, the plasmid pGNT-*lcsG* and the empty vector pGNT were transformed into PLBJ-1 to construct the overexpression and control strains, respectively.

### qRT–PCR analysis of the *lcsG* overexpression strain

For cDNA synthesis, approximately 1 μg of DNase-treated, RNase-cleaned RNA was used as the template by using HiScript III RT SuperMix for qPCR (+gDNA wiper) (R312, Vazyme Biotech Co., Ltd, China). Three biological replicates were measured for each analysis of the relative expression levels. The housekeeping gene *actin* (GenBank number VFPBJ_07912) was used as a control. qRT–PCR was performed with ChamQ Universal SYBR qPCR Master Mix (Q711, Vazyme Biotech Co., Ltd, China) on a Bio-Rad CFX96 (Bio-Rad). The relative expression values were calculated using the $2^{-\Delta\Delta Ct}$ method[60]. The primers used are listed in Supplementary Table 9.

### Culture extraction

Cultures of $1 \times 10^5$ conidia per mL of PLBJ-1 and its mutants were cultured in PDB media at 28 °C and 220 rpm for 14 days. The fermentation mixture was extracted with an equal volume of ethyl acetate (EtOAc) three times (every 1 h) to efficiently extract leucinostatins and EtOAc was evaporated under reduced pressure. The extract was redissolved in acetonitrile (MeCN) for further experiments.

### Product purification of LeuK0, LeuA0, and LeuK3

LeuK0, LeuA0, and LeuK3 were purified by semipreparative HPLC from the crude extracts mentioned above. The UV absorption of leucinostatins was monitored at 214 nm with HPLC DAD. The samples were separated on an Agilent 1260 Infinity II HPLC system with a Kromasil 100-5-C18 column (10 mm × 250 mm), and eluted with a linear gradient of MeCN-water, starting at 20% MeCN and reaching 70% MeCN over 25 min at a flow rate of 2 ml/min. The retention times of LeuK0, LeuA0, and LeuK3 were 23.8, 21.6, and 22.3 min, respectively. LeuK0 (18 mg, purity ≥95%) was obtained from the crude extract of Δ*lcsG* mutant (570 mg). LeuA0 (13 mg, purity ≥95%) and LeuK3 (11 mg, purity ≥95%) were isolated from chemically methylated WT (390 mg) and Δ*lcsG* mutant crude extracts (500 mg), respectively.

### Chemical methylation of LeuA and LeuK0

For the chemical methylation pilot study of LeuA and LeuK0, diisopropylethylamine (15 μl) and iodomethane (55 μl) were successively added to solutions of LeuA and LeuK0 (10 mg) in dry tetrahydrofuran (THF) (0.6 ml), respectively. These mixtures were stirred at room temperature at 800 rpm for 46 h, and volatile constituents were evaporated at room temperature by using nitrogen blowdown evaporator[61]. To isolate LeuA0 and LeuK3 in bulk, the WT and Δ*lcsG* mutant crude extracts were used as substrates, respectively.

### Structural characterization of LeuK0

LeuK0 was assigned a molecular formula of $C_{62}H_{111}N_{11}O_{14}$ based on its HRESIMS data ($m/z$ 1234.8383 $[M + H]^+$). ESI-MS-MS data were compared with those of previously reported leucinostatins A–C, revealing identical structural features (from B1 to C10) but a distinct C-terminal unit ($C_2H_6NO$) in LeuK0. Analysis of its $^{13}C$-NMR APT, DEPT-135, and DEPT-

90 spectroscopic data revealed a total of 62 carbons, including 18 methyl groups (-CH$_3$), 14 methylenes (-CH$_2$), 16 methines (-CH), and 14 sp$^3$ quaternary carbons. Since the known unit (from B1 to C10) already contains 18 -CH$_3$, 12 -CH$_2$, 16 -CH, and 14 sp$^3$ quaternary carbons, the C-terminal unit (C$_2$H$_6$NO) in LeuK0 was thought to be -NH-CH$_2$-CH$_2$-OH or -CH$_2$-CH$_2$-O-NH$_2$.

## *N*-hydroxysuccinimide (NHS)-ester reaction

7-methoxycoumarin-3-carboxylic acid N-succinimidyl ester (4 mg) and diisopropylethylamine (6 µl) were successively added to a solution of LeuK0 (15 mg) in dimethylformamide (DMF) (200 µl). These mixtures were stirred at room temperature at 800 rpm for 3 h, and the volatile constituents were evaporated at room temperature by using nitrogen blowdown evaporator.

## Protein expression and purification

For the expression of LcsG, *E. coli* ArcticExpress (DE3) carrying pACYC-*lcsG* was cultured. The *E. coli* cells were grown at 37 °C in 1 l of LB medium supplemented with the appropriate antibiotics. The culture was supplemented with IPTG (at a final concentration of 0.1 mM) when the OD$_{600}$ reached 0.6–0.8, after which the induced *E. coli* were grown at 11 °C for 24 h. The cells were harvested by centrifugation (5000 rpm, 15 min, 4 °C), resuspended in 20 ml of lysis buffer (50 mM NaH$_2$PO$_4$, pH 8.0, 300 mM NaCl, 10 mM imidazole) and lysed by sonication on ice (200 W, 10 s, 10 s, 20 min) with an ultrasonic homogenizer SCIENTZ-IID (SCIENTZ, China). The lysate was centrifuged (12,000 × *g*, 1 h, 4 °C), and the supernatant was collected and filtered through a 0.45 µm membrane (Sartorius Stedim Biotech, Germany) to remove residual cellular debris. A nickel affinity chromatography column was prepared with 2 ml of Ni-NTA resin (Trans Gen Biotech, China) loaded into a gravity column (Sangon, China) and used to separate the filtered supernatant. After the samples were washed with wash buffer (50 mM NaH$_2$PO$_4$, pH 8.0, 300 mM NaCl, and 20/40/60 mM imidazole), His-tagged proteins were eluted with elution buffer (50 mM NaH$_2$PO$_4$, pH 8.0, 300 mM NaCl, and 250 mM imidazole). Purified proteins were concentrated in storage buffer (50 mM Tris-HCl, pH 7.5, 20% glycerol), and flash frozen. The purity of the protein was confirmed by SDS-PAGE. The mutant proteins were purified via a procedure similar to that used for LcsG. The expression of the mutated proteins was verified by Western blotting using an anti-His antibody (TransGen Biotech Ltd., Beijing, China).

## In vitro enzyme assay for LcsG and its mutants

For the benchmark experimental conditions, the enzyme assays (50 µl) contained Tris-HCl buffer (50 mM, pH 7.5), SAM (2 mM), LeuA or LeuK0 (10 µM), and purified recombinant LcsG (10 µM). The reactions were incubated at 28 °C for 120 min. For single-factor experiments, the enzyme assays (50 µl) contained Tris-HCl buffer (50 mM), SAM (2 mM), PLBJ-1 WT crude extract (200 µg), and purified recombinant LcsG (10 µM). For the reaction time course assay, LcsG was tested with Tris-HCl buffer (pH 7.5) at 28 °C for 0.01, 15, 30, 45, 60, 75, 90, 105, and 120 min. For the reaction pH course assay, LcsG was tested at 28 °C for 120 min with Tris-HCl buffer at different pH values (pH 7.0, pH 7.5, pH 8.0, pH 8.5). For the reaction temperature course assay, LcsG was tested with Tris-HCl buffer (pH 8.0) for 120 min at 18, 21, 27, 34, and 38 °C. To determine the function of mutated LcsG, 50 µl reaction mixtures were prepared with 500 nM mutant catalysts, 2 mM SAM, 100 mM Tris-HCl buffer (pH 8.0), and 10 µM substrates. The reactions were incubated at 34 °C for 120 min. The above enzymatic reactions were quenched by adding 50 µl cold acetonitrile and 10 µl was analyzed by LC-MS.

## Calibration curves of LeuA0 and LeuK3

The compounds were quantified by an external standard method and calibration curves were constructed using LeuA0 and LeuK3. In this study, LeuA0 was diluted to concentrations of 25, 50, 100, 250, and 500 nM. LeuK3 was also diluted to concentrations of 50, 100, 250, 500, and 1000 nM. Dilutions were prepared in MeCN, and calibration curves were constructed using the software GraphPad Prism 9.

## Michaelis–Menten enzyme kinetics

The kinetic constants of LcsG were determined by the following approach: 1 µM LcsG was assayed against different specific concentrations of the substrates LeuA (2, 4, 8, 10, 20, 40, 80, 100, 120, 150, and 200 µM) and LeuK0 (4, 8, 10, 20, 40, 80, 100, 120, 140, and 160 µM). The initial rate of the reaction was measured by monitoring the formation of products (LeuA0 or LeuK3) at 34 °C for 30 min. The rates were plotted against substrate concentration using the Michaelis−Menten kinetics equation by nonlinear regression analysis with the software GraphPad Prism 9, and $K_m$ and $K_{cat}$ constants were generated from the resulting Michaelis−Menten plot.

## LC–MS analysis

LC−MS analyses were performed on an Agilent 1290 Infinity II HPLC with an Agilent Infinity Lab single quadrupole mass selective detector by using an Agilent Zorbax Eclipse Plus C18 reversed-phase column (2.1 × 100 mm, 2.7 µm). Water (A) with 0.1% (v/v) formic acid and acetonitrile (B) were used as the solvents at a flow rate of 0.25 ml min$^{-1}$. The substances were eluted with 10% (v/v) B for 1 min, subjected to a linear gradient from 10 to 100% (v/v) B for 12 min, washed with 100% solvent B for 5 min, and equilibrated with 5% solvent B for 10 min at a flow rate of 0.25 ml/min. The mass spectrometer was set in electrospray positive ion mode for ionization.

LC-HRESI-MS-MS analyses were performed on an Agilent HPLC 1260 Infinity II system equipped with an Agilent G6510A mass spectrometer by using an Agilent Zorbax SB-C18 reversed-phase column (4.6 × 150 mm, 5 µm). A linear gradient analytical method was used (10–100% MeCN in water with 0.1% formic acid for 20 min at a flow rate of 1.0 ml/min). The Q-TOF was operated in positive electrospray ionization mode with a capillary voltage of 1800 V and a drying gas flow rate of 1 µl/min at 300 °C. The MS scan range was 80–2000 *m/z*, and the MS-MS scan range was 40–1400 *m/z*. The fixed collision energy was 65 V.

## Microbial growth inhibition assays

The growth inhibition of *C. neoformans* H99 by leucinostatins (LeuK0, LeuK3, LeuA0) with was assessed on PDA media with agar diffusion assays. Overnight cultures of *C. neoformans* grown in PDB broth at 28 °C were diluted with PDB broth to an OD$_{600}$ of 0.1. One milliliter aliquots of the resulting mixture were combined with 30 ml aliquots of PDA at 45 °C. The test wells (4 mm diameter) were aspirated from the solidified medium using the tip of a sterilized Luer-lock syringe, and 10 µl of each compound (25 µg) was added to each well. The plate was incubated at 28 °C. Zones of inhibition were photographed after 36 h, and leucinostatins and amphotericin B (Solarbio, China) were dissolved in DMSO. The growth inhibition of *P. infestans* by leucinostatins (LeuK0, LeuK3, LeuA0) with was assessed on rye agar medium in 9-cm Petri plates. *P. infestans* was incubated on the center of plates and cultured at 18 °C for 3 days, followed by incubation in aspiration test wells (4 mm diam) at the colony edges.

According to the National Committee for Clinical Laboratory Standards (NCCLS) recommendations[62], the minimal inhibitory concentration (MIC) was determined with three replicates using the serial dilution method in 96-well plates with YM (1% maltose extract, 0.2% yeast extract) as the test medium. Amphotericin B was used as the positive control. Test compounds were dissolved in DMSO and serially diluted in a growth medium. The visual endpoint and optical density of the microplate wells were measured relative to those of the positive and negative controls. The *C. neoformans* H99 strains were incubated at 25 °C, and the MICs were determined at 48 h. Viability was determined with the aid of a plate reader using PrestoBlue resazurin dye (Life Technologies) as the viability indicator. The spectrophotometric MIC value was defined as the lowest concentration of a test compound that resulted in a culture with a density equal to 100% inhibition compared to the growth of the untreated control.

## Structure prediction of LcsG

Uni-Fold (https://github.com/dptech-corp/Uni-Fold) was used to predict the LcsG dimer structure. A fasta file of two lcsG sequences was uploaded, and a predicted structure was returned. The average pLDDT score was 0.88.

## Structure and sequence alignment

Homologous protein structures were searched via HHpred. ChimeraX was used to estimate the structural alignment and plot the figures. Multiple sequences were aligned by MAFFT, and ESpript3 (https://espript.ibcp.fr/ESPript/cgi-bin/ESPript.cgi) was subsequently used to export the sequence alignment results.

## DiffDock

The structure of leucinostatin A was modified from the structures of its analogs ZHAWOC6027(PDB: 8a19) and helioferin A (PDB: 6evh). The DiffDock webserver (https://huggingface.co/spaces/simonduerr/diffdock) was used to simulate the dock. The predicted LcsG dimer structure, and leucinostatin A structure were uploaded to initiate the docking process, and the website's backend algorithms were used to process the data and predict the ligand's binding poses to the protein. The model was then chosen from the final docking poses as well as the top-ranked docking pose.

## Molecular dynamic simulation

Molecular dynamics simulations were performed using Desmond (Schrödinger, LLC, NY, USA)[63]. The protein-ligand complex was used as an input for the system builder. The OPLS4 force field was employed, the water model used was SPC, and the concentration of NaCl was 0.15 M. The final system was composed of a protein-ligand complex, 15062 water, 53 $Na^+$ and 42 $Cl^-$. The output of the system builder was then loaded into the Energy Minimization panel. After energy minimization, the system was then loaded into the Molecular Dynamics panel. The simulation was carried out with the default NPT run settings in Desmond; the temperature was 300 K, the pressure was 1 bar and the simulation time was 500 ns. Later, we extract the last frame of the simulation and run 3 more independent 500 ns simulations.

## Bioinformatics analysis of LcsG and phylogenetic tree construction

In total, 130 sequences of methyltransferases with annotated functions were downloaded from the NCBI database. The conserved domain architecture of these methyltransferases was characterized using the SMART domain prediction tool[64] (http://smart.embl-heidelberg.de/). The specific parameters for domain prediction were set to include the PFAM and SMART domain databases. For phylogenetic analysis, a maximum likelihood tree was estimated utilizing IQ-TREE software version 1.6.12[65] with a bootstrap of 1000. The resulting tree was visualized using the Interactive Tree of Life (ITOL, http://itol.embl.de/).

## Sequence similarity networks

A search for sequences similar to LcsG was performed using the NCBI BLASTp tool, setting the expectation threshold at 0.05. This search specifically identified sequences from fungi within the same Pfam (PF00891) with LcsG that have been demonstrated by experiments. These selected sequences were then analyzed using the EFI-EST enzyme similarity tool[66]. A sequence similarity network (SSN) was constructed for the domains of LcsG and these related sequences. For visualization purposes, an alignment score cutoff of 10 was employed, and sequences with ≥40% identity were combined into representative nodes. This SSN was subsequently visualized using Cytoscape version 3.9.1.

## Statistics and reproducibility

Every experiment was conducted with a minimum of three independent biological replicates. Data are presented as the mean ± standard deviation (SD). Data points in figures represent biological replicates. Each graph displays individual data points. Statistical analyses (mean and SD) were performed using GraphPad Prism 9 software. Detailed information on all reagents and resources can be found in "Methods" section.

## Reporting summary

Further information on research design is available in the Nature Portfolio Reporting Summary linked to this article.

## Data availability

Uncropped images of Fig. 2a and Supplementary Figs. 2 and 24b are provided in Supplementary Fig. 29. Information of experimentally confirmed methyltransferase analogous to LcsG used for phylogenetic tree construction in this study is in Supplementary Data 1. The source data behind the graphs are in Supplementary Data 2. All initial coordination and simulation input files of the MD simulations are in Supplementary Data 3. Any remaining information can be obtained from the corresponding author upon reasonable request.

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

## Acknowledgements

This study was financially supported by grants from the National Key R&D Program of China (2023YFD1400400 and 2022YFD1400700), the National Natural Science Foundation of China (32272630) and the Agricultural Science and Technology Innovation Program of CAAS. We thank Prof. Junfeng Liu and Dr. Xin Zhang (China Agricultural University, China) for their advice on protein purification. We appreciate the support from Nanjing University of Chinese Medicine and the provision of the Maestro software by Schrödinger. We also thank the company DP Technology, China, for providing advice on protein structure prediction and molecular dynamic analysis.

## Author contributions

Y.L., B.X., and Z.L. designed the research. Z.L. performed protein purification, fungal fermentation, compounds isolation, structure elucidation, LC-MS analysis, in vivo genetic and in vitro biochemical experiments; Y.J. performed protein purification, LC-MS analysis, in vivo genetic and in vitro biochemical experiments; J.L. and J.Z. performed the genomic analysis; Z.M. and Y.Y. assisted in the test of antimicrobial activity; K.Z. assisted in structure elucidation and chemical synthesis; W.W. performed LC-MS analysis and LC-HRESI-MS-MS analysis; Y.L., B.X., and Z.L. wrote the manuscript.

## Competing interests

The authors declare no competing interests.
