## [Peer review file · Communications Biology]

Reviewers' comments:

Reviewer #1 (Remarks to the Author):

N-Methyltransferase (NMT)-catalyzed methylations at nonribosomal peptide (NRP) termini are uncommon, making them an intriguing area of research. The authors discovered LcsG, a fungal N-methyltransferase that is responsible for iterative terminal N-methylation of a family of NRPs known as leucinostatins produced by *Purpureocillium lilacinum*. They demonstrated the importance of LcsG in the methylation of leucinostatins using gene deletion experiments. They used *in vitro* assays and HRESI-MS-MS analyses to determine the specific sites of methylation. These experiments demonstrated conclusively that the C-terminal unit of various leucinostatins is methylated at the NH₂, NHCH₃, and N(CH₃)₂ positions. Furthermore, they used molecular docking and site-directed mutagenesis to formulate a hypothesis about the catalytic mechanism of LcsG by leveraging the predictive capabilities of artificial intelligence (AI). The model proposes that an N atom in LcsG is coordinated by two negatively charged residues, Asp368 and Asp395, facilitating a subsequent SN₂ methylation reaction.

These findings not only point to a promising strategy for broadening the natural bioactivity of NRPs, but they also shed light on the complex catalytic mechanism that governs N-methylation in NRPs. However, in its current form, the manuscript leaves a lot of room for improvement.

Here are my specific comments:

Major comments:

The title should be more specific and reflect the content of the study, focusing on leucinostatins and their N-methylation.

The introduction should provide important background information on nonribosomal peptides (NRPs), their significance, and the importance of N-methylation. This section should also discuss leucinostatins and their relevance to NRPs. The need for studying NMTs in the context of leucinostatins should be clearly defined.

Arrange the results section logically, beginning with the identification of the critical function of LcsG in leucinostatin methylation and proceeding to the *in vitro* assay outcomes and the HRESI-MS-MS examination. It is necessary to include a section in the results section to assess the biological activities of the novel leuconostatins.

To help with reproducibility, provide clear experimental details such as parameters, conditions, and protocols. A good number of sections appear to be lacking important details.

Minor comments:

Page 2, lines 20-21: Change ...N-methyl formation of a family of NRPs... to ...N-methylation of a family of NRPs...

Page 2, line 23: Change ...the C-terminal unit of... to ...the C-termini of...

Page 2. Lines 25-27: Consider rephrasing the sentence to make it clearer: ...we proposed the catalytic mechanism of the LcsG-catalyzed reaction was an N atom coordinated by two negatively charged residues (Asp368, Asp395 for LcsG) towards the subsequent SN2 methylation...

Page 3. Lines 40-41: The study's unique contribution is the iterative N-methylation, which the author should highlight in the introduction and go into more detail about.

Page 3, line 47: Change ...a NRPS... to ...an NRPS...

Page 3, line 48: Here and in other places in the manuscript and SI section, change the C in C-terminus to a non-italic character.

Page 4, lines 54-55: Change ...we identify a discrete NMT that could catalyze a unique moiety located in the terminus of a NRP, named as LcsG, from the biological control fungus... to ...we identify a discrete NMT, named as LcsG, which could catalyze a unique moiety located in the terminus of a NRP from the biological control fungus...

Page 4, line 58: Change ...demonstrate... to ...demonstrated...

Page 4, lines 67-70: To make the sentence easier to read, it needs to be rephrased.

Page 4, line 73: How do OE1 and OE2 in Fig. S3 relate to OE_{LcsG}? Clarify this in this passage of text.

Page 5, lines 81-82: In spite of the fact that peak 4 Leu was a methylation derivative of Leu_C, the authors should explain why it was given the name K₀ rather than C₀.

Page 5, lines 85-87: The deletion of the *lcsG* gene eliminated Leu A and B but not Leu_C. Why? This should be explained by the authors.

Page 5, line 92: Indicate whether the 6xHis tag was at the N or C terminus.

Page 5, lines 93-96: What was the reasoning behind this substrate selection? Why wasn't Leu_B tested? And how did Lue_{K0} come to be tested in this set of experiments? The way the text progresses makes it difficult to understand this.

Page 6, lines 103-5: Once again, determining the source of the substrate is difficult. The authors must make it abundantly clear what the substrate was and where it came from, as well as that WT refers to the fungus. This may be clarified by referring to the following section.

Page 6, lines 112-14: How the precise substrate concentrations were adjusted in this range. Authors must elaborate in the text. Again, the authors can avoid confusion by adequately cross-referencing the methods section.

Page 6, lines 114-16: This sentence appears to be missing text or has a typo, and thus needs to be corrected and rephrased.

Page 7, lines 130-40: This section should be rewritten to be more concise and readable.

Page 8, line 55: Change ...we employ... to ...we employed...

Page 9, lines 169-70: Add a bit more on mutating the five additional Asp residues (SAM binding) of LcsG to Ala in addition to the ones flanking and close to the N-terminus of LeuA to be methylated.

Page 9, lines 172-74: Rewrite the sentence to make it easier to read.

Page 10, line 194: A separate section in the results section should be added to evaluate the biological activities of new leucinostatins.

Page 15, lines 309-11: How did these 10 mg of LeuA and K0 come about? How were other Leu substrates purified and quantified before use in assays? This should be clearly described in the methods section. The problem is that this information is not covered in the preceding sections, and there is no cross-referencing to the following sections.

Page 15, line 314: To make sense and avoid confusion regarding the use of purified substrate in the assays, this section must come before the previous section.

Page 15, lines 315-18: These details, as well as the quantities of purified leucinostatins obtained and their purity levels, should be included in the supplementary file.

Page 17, line 364: Change ...time cause assay... to ...time course assay...

Page 17, lines 365-66: Change ...pH cause assay... to ...pH assay...

Page 17, line 367: Change ...temperature cause assay... to ...temperature assay...

Page 18, line 370: Change ...catalysts... to ...mutant catalysts...

Page 18, line 380: This range is stated in the results as 0-200 μ M. State the concentration ranges in a consistent and uniform manner.

Page 19, line 406: What was the concentration of test compounds in the 10 μ L solutions added to the plate wells?

Page 19, lines 409-10: Provide specifics about the ingredients of the rye agar that was used.

Page 19, line 411: What was added to the wells that were aspirated is unclear. How the inhibition zone was determined?

Page 20, line 418: Add the strains that were used.

Page 20, lines 412-22: This section needs to be rewritten so that the strains being tested are made crystal clear. It needs to be rephrased because it doesn't read well as it is.

Page 20, line 423: This needs to be expanded to include information on how the predicted structure was obtained using the Uni-fold. Here, more information about the step-by-step process is required.

Page 20, line 430: For repeatability, this section needs to be updated with enough details. Here, too, additional details regarding the methodical procedure are needed.

Page 31, line 602: For Fig. 4D, specify the assay, sample size, and statistics.

SI, Page 20: Give the sample size and the reason behind the absence of SD error bars in Fig. S13.

Reviewer #2 (Remarks to the Author):

This manuscript presents the first characterization of an unusual N-methyltransferase, LcsG, implicated in the biosynthesis of leucinostatins, a family of nonribosomal peptides known for their potent bioactivity. Through a series of biochemical assays, the authors propose that LcsG is a separately encoded modifying-enzyme that acts in trans to the non-ribosomal peptide synthetase machinery to sequentially methylate the C-termini of the associated natural products. They additionally identify a series of novel leucinostatins with increased antifungal properties. While these results are certainly intriguing, the importance of these contributions is not clear, and I was not convinced by the mechanistic arguments. Furthermore, the text does not read wonderfully smoothly with many concepts referenced before they are defined, awkward transitions between topics, and a variety of grammatical/typographical errors.

With regards to impact, the authors state that “NMT-catalyzed iterative N-methylation at the terminus of NRPs has been rarely reported.” However, there is no reference or context provided here. Has it been reported previously, or not, to the best of the authors’ knowledge? Without this clarification, the novelty of the described reactions is difficult to assess.

Likewise, on page 7, the authors reference leucinostatins as well-known antibiotics, but there are no references here. They then proceed to discuss their antifungal assays. Why perform antifungal assays if these molecules are commonly antibiotics? Are the molecules in this study also antibiotics? Are their antifungal properties unusual for the family of NPs? Why were the particular strains selected for testing in the first place? How do the MICs compare to other relevant molecules? Clarification is needed here to place the presented results in context with the broader scientific literature in this area.

The least compelling aspect of the manuscript was the mechanistic hypothesis, which primarily relies on the AI-predicted structure and subsequent docking simulations. As the structure is predicted, not experimental, it is important to address the confidence of the model. It is noted in the Methods section that the average pLDDT score was 0.88. While this is certainly reassuring, it does not provide insight into what parts of the structure were lower confidence. If active site loops/residues are lower confidence, then the confidence in the docking models is equally questionable. A discussion of the pLDDT surrounding the active site is warranted. A figure of the overall structure colored by pLDDT may also be useful depending on the distribution of uncertainty. Along these same lines, were any residues considered flexible in the

docking simulations? If the AI model was less confident about particular residues/sidechains, these should not be static in the docking simulation.

Why was only the top ranked docking pose considered? How different was this pose to the next highest ranked pose? What was the difference in the ranking? These rankings are far from perfect. Did the authors consider any single point energy calculations or short MD simulations to assess stability of the poses to ensure an optimally modeled interaction. This seems especially relevant given attempts to infer mechanistic behavior. A figure depicting the distances between the putative catalytic residues, the targeted position, and the SAM methyl in the proposed model would help to motivate these discussions.

Furthermore, while mutagenesis studies highlight the importance of D368 and D395 in catalysis, these results do not provide an airtight case for their involvement in the deprotonation step. Are there other residues nearby that could also potentially serve in this capacity? Is it possible one or both residues are solely important for binding? How many other interactions hold the substrate in place? Does a double mutant abolish activity?

More specific questions/critiques are detailed below.

The authors state that “the deletion of lcsG only led to the abolishment of LeuB and LeuA, suggesting...” However, there still appears to be a small peak in Figure 1c around the elution time of LeuA in the trace for the LcsG knockout. Is this something else? This should be clarified.

The experiments to assess kinetics on page 6 were very confusing. The authors state that they used “the crude extract of WT as the substrate”?? The accumulation of various products is then described without a clear description of the experiments. The Fig. 2c caption is equally unhelpful. Was only 2 or 4 provided at the start of the reaction, or were both 2 and 3, or 4 and 6 provided for these assays? How can the peak areas for what I presume are the selected substrates for each reaction be negative? While I can see the ‘slight increase’ the authors note for 3, I do not see the same for 6 in the Fig. 2c data.

Whenever structural alignments are discussed/depicted, the authors should not the RMSDs. Are these alignments of just the backbone? How many atoms were included, etc.? These values are important for gauging similarity.

The author’s state that accurate stereochemistry of leucinostatins is hard to predict before basing their structure of LeuA off the structures of similar molecules. How confident are they in this stereochemical assignment? Did they try docking alternatives?

Did the authors consider generating a SSN as well a phylogenetic tree? This approach may provide additional insights into the relationship between these enzymes or help to identify other similar systems.

In the discussion, the authors compare the catalytic residues of OxaC and CHOMT to LcsG. This comes out of nowhere. Why reference these two specific enzymes? While Fig. S22 purportedly corresponds to this discussion, I don’t see these two enzymes in that sequence alignment...?

Other figure comments

Figure 1, panel c – What is the top smaller panel here? Is it a zoomed-in view of the first peak? While the position of the peak on the x-axis appears to suggest it is not (i.e. the peaks in the top spectrum tail off around 15.6 min, while those in the bottom panel tail off around 16 min), this is very confusing.

Compounds 6-8 are referenced in the text and in Fig. 2 before they are even introduced. They are not described until the reader gets to Fig. 3. To avoid confusion, a figure of these compounds needs to be presented earlier in figure or text.

Currently, there are no figures of the full AI prediction of the overall enzyme structure. Such an image would allow the authors to label the different domains referenced in the text. Furthermore, a comparison with other related NMTs and OMTs would be very helpful to provide a visualization of the overall similarities or lack thereof.

Comparison between OMT and NMT mechanisms was exceedingly difficult without schemes for reference. The sequence alignments of Fig. S17, S19 and S22 were also poorly captioned, difficult to correlate with the discussion in the text.

Generally speaking, the resolution of figures in the SI is problematic. This is particularly true for S9 and S14, which are difficult to read both due to their size and resolution.

The sequence alignment in Fig. S17 contains a series of boxed, highlighted, marked residues. However, there is no indication in the caption as to what the red arrows, red highlighting and blue boxes are pointing out exactly. Either a lengthier caption, or legend is warranted here. It would be helpful to label the SAH/SAM binding motif, proposed catalytic residues, etc. This figure should also be referenced in the text when discussing these features.

There is confusing text throughout. A couple of selected examples are listed below, but these are scattered throughout contributing to confusion. Perhaps it would be worth investigating an editing service. Note that the plural of C-terminus is C-termini not C-terminuses.

- “However, the biosynthetic mechanisms of the diverse C-terminuses remain unknown.” I presume the authors meant the biosynthetic mechanisms to generate the diverse C-termini as the C-termini don’t have mechanisms?

- The authors write “In this work, we identify a discrete NMT that could catalyze a unique moiety located in the terminus of a NRP, named as LcsG, from the biological control fungus *P. lilacinum* PLBJ-1.23”

“catalyze a unique moiety” doesn’t make sense.

As is, the text implies LcsG is the NRP, but it is the modifying enzyme.

This would read more smoothly as follows: “In this work, we identify a discrete NMT, LcsG, that catalyzes the installation of a unique moiety located at the terminus of an NRP from the biological control fungus *P. lilacinum* PLBJ-1.23”

Characterization of *N*-methyltransferase for catalyzing the terminus of leucinostatins in *Purpureocillium lilacinum*

Reviewer #1

1) The title should be more specific and reflect the content of the study, focusing on leucinostatins and their *N*-methylation.

Response: We have modified the title as: “Characterization of a methyltransferase for catalyzing iterative *N*-methylations at leucinostatin lipopeptides termini”

2) The introduction should provide important background information on nonribosomal peptides (NRPs), their significance, and the importance of *N*-methylation. This section should also discuss leucinostatins and their relevance to NRPs. The need for studying NMTs in the context of leucinostatins should be clearly defined.

Response: We have expanded introduction with comprehensive NRP background, emphasized *N*-methylation's role in NRPs, highlighted leucinostatins' relevance in NRPs, and clarified why NMTs are studied in leucinostatins. Please see the introduction section in page 3 lines 35-58.

3) Arrange the results section logically, beginning with the identification of the critical function of LcsG in leucinostatin methylation and proceeding to the *in vitro* assay outcomes and the HRESI-MS-MS examination. It is necessary to include a section in the results section to assess the biological activities of the novel leuconostatins.

Response: In response to your guidance, we have restructured the results section for enhanced coherence. This article contains 5 results: identification of LcsG from *Purpureocillium lilacinum* by gene deletion, *in vitro* assay, products elucidation by HRESI-MS-MS, antifungal evaluation of leucinostatins, and catalytic mechanism hypothesis. “the biological activities of the novel leuconostatins” part has been addressed in the text, please see the results section “antifungal evaluation of leucinostatins” in page 8 lines 170-178.

4) To help with reproducibility, provide clear experimental details such as parameters, conditions, and protocols. A good number of sections appear to be lacking important details.

Response: We have addressed the concern by including additional experimental details, encompassing the parameters, conditions, and protocols as suggested. Please refer to the yellow highlighted parts in the methods section.

5) Page 2, lines 20-21: Change ...*N*-methyl formation of a family of NRPs... to ...*N*-methylation of a family of NRPs...

Response: “...*N*-methyl formation of a family of NRPs...” has been changed to “...*N*-methylation of a family of NRPs...”.

6) Page 2, line 23: Change ...the C-terminal unit of... to ...the C-termini of...

Response: “...the C-terminal unit of...” has been changed to “...the C-termini of...”

7) Page 2. Lines 25-27: Consider rephrasing the sentence to make it clearer: ...we proposed the catalytic mechanism of the LcsG-catalyzed reaction was an N atom coordinated by two negatively charged residues (Asp368, Asp395 for LcsG) towards the subsequent SN2 methylation...

Response: We have revised it to “we proposed that the catalytic mechanism of the LcsG-catalyzed reaction involves the coordination of an N atom by two negatively charged residues (Asp368 and Asp395 in the case of LcsG), which facilitates the subsequent SN2 methylation”

8) Page 3. Lines 40-41: The study's unique contribution is the iterative *N*-methylation, which the author should highlight in the introduction and go into more detail about.

Response: We have changed “MT-catalyzed iterative *N*-methylation at the terminus of NRPs has been rarely reported.” to “To our knowledge, a discrete NMT-catalyzed iterative *N*-methylation is not previously observed in the terminus of NRPs”. We also discussed the rare occurrence of iterative NMT for catalyzing multiple methyl transfers in the discussion section first paragraph.

9) Page 3, line 47: Change ...a NRPS... to ...an NRPS...

Response: “...a NRPS...” has been changed to “...an NRPS...”

10) Page 3, line 48: Here and in other places in the manuscript and SI section, change the C in C-terminus to a non-italic character.

Response: We have changed all of C in C-terminus to a non-italic character in the manuscript and SI section.

11) Page 4, lines 54-55: Change ...we identify a discrete NMT that could catalyze a unique moiety located in the terminus of a NRP, named as LcsG, from the biological control fungus... to ...we identify a discrete NMT, named as LcsG, which could catalyze a unique moiety located in the terminus of a NRP from the biological control fungus...

Response: “...we identify a discrete NMT that could catalyze a unique moiety located in the terminus of a NRP, named as LcsG, from the biological control fungus...” has been changed to “...we identify a discrete NMT, named as LcsG, which could catalyze a unique moiety located in the terminus of a NRP from the biological control fungus...”

12) Page 4, line 58: Change ...demonstrate... to ...demonstrated...

Response: “...demonstrate...” has been changed to “...demonstrated...”

13) Page 4, lines 67-70: To make the sentence easier to read, it needs to be rephrased.

Response: We have changed that sentence to “LcsG is equipped with the Methyltransf₂ domain, which is commonly found in *O*-methyltransferases (OMTs) (Table S1). The top hit in its pBLAST search shared 31.65% sequence identity with an OMT called VdtC (A0A443HJY8.1). However, despite this prediction, we were unable to identify any *O*-methylated units in leucinostatins. This inconsistency aroused our curiosity and prompted us to investigate the function of LcsG.” Please see in page 5 lines 85-89.

14) Page 4, line 73: How do OE1 and OE2 in Fig. S3 relate to OE_{LcsG}? Clarify this in this passage of text.

Response: We have revised that to “To figure out the function of LcsG in the leucinostatins biosynthesis, we constructed deletion mutants ($\Delta lcsG$) and overexpression mutants (OE $lcsG$) of *P. lilacinum* PLBJ-1 (Fig. S1-S3). Following growth on a productive medium and production extraction, LC-MS analysis suggested obvious differences between the deletion mutant and wild-type strain (WT) (Fig. 1C), while there were no obvious differences between the overexpression mutants (OE1, OE2, which correspond to strains overexpressing the *lcsG* gene) and WT (Fig. S3).” Please see in page 5 lines 90-95.

15) Page 5, lines 81-82: In spite of the fact that peak 4 Leu was a methylation derivative of LeuC, the authors should explain why it was given the name K0 rather than C0.

Response: Based on the same molecular weight and HRESI-MS-MS results, peak 4 was initially presumed to be leucinostatin K (LeuK), a compound isolated from *Paecilomyces lilacinus* (synonym *Purpureocillium lilacinum*). However, the NMR analysis suggested peak 4 was not the known compound LeuK but a compound derived from LeuC. As its structure was not fully consistent with LeuC and had not been reported before, and the name “K0” was unclaimed, we designated peak 4 as leucinostatin K0 (4, LeuK0) mainly based on that the peak 4 exhibited a molecular weight and HRESI-MS-MS results consistent with the LeuK Please see the section in page 5 lines 99-105.

16) Page 5, lines 85-87: The deletion of the *lcsG* gene eliminated Leu A and B but not Leu C. Why? This should be explained by the authors.

Response: This difference in methylation at the C-terminus is attributed to the presence of methylated C-termini in LeuA and LeuB, while LeuC lacks this modification. Please see the section “The deletion of *lcsG* only led to the abolishment of LeuB and LeuA, both of which have methylated termini, suggesting the deletion blocked the formation of the methylated C-terminal amines” in page5 lines 108-110.

17) Page 5, line 92: Indicate whether the 6xHis tag was at the N or C terminus.

Response: We have indicated it as “...we expressed *lcsG* in *E. coli* ArcticExpress (DE3) and purified the recombinant protein LcsG, which was tagged with His₆ at both the N and C termini, using nickel affinity chromatography...”

18) Page 5, lines 93-96: What was the reasoning behind this substrate selection? Why wasn't LeuB tested? And how did LueK0 come to be tested in this set of experiments? The way the text progresses makes it difficult to understand this.

Response: Obtaining pure LeuA, LeuB or LeuC in laboratory is exceptionally hard. LeuA is commercially available while LeuB and LeuC are not. Through the HPLC-MS, we observed distinct signal peaks for LeuB and LeuA in the metabolic products of the PLBJ-1 strain (Fig 1C). However, despite of many attempts, when conducting HPLC analysis or performing purification processes, it has been extremely difficult to achieve adequate separation of LeuA, LeuB, and LeuC. The challenge of separating LeuA and LeuB for HPLC could also be corroborated by findings in Wang's article (Wang G, Liu Z, Lin R, et al. Biosynthesis of antibiotic leucinostatins in bio-control fungus *Purpureocillium lilacinum* and their inhibition on *Phytophthora* revealed by genome mining[J]. PLoS pathogens, 2016, 12(7): e1005685.)" as depicted in Figure 5. Following the observation that the deletion of *lcsG* led to the predominant presence of LeuK0 in the metabolites, we aimed to investigate whether catalyzing the $\Delta lcsG$'s metabolites could restore

the productions of WT strain. As we depicted in Fig. 1C, the main component from $\Delta lcsG$ mutant is LeuK0 and a minimal amount of LeuC. Thus LeuK0 could be well separated from the crude extract of $\Delta lcsG$ mutants. We initially conducted preliminary experiments using this crude extract and detected the monomethylated product LeuK2 and dimethylated product LeuK3 of LeuK0. This result provides initial evidence for the methyltransferase activity of LcsG. In the formal experiments, we utilized purified LeuK0 as the substrate and confirmed the presence of same methylation products. Additionally, another assay using commercial LeuA also revealed the trimethylated product LeuA0. Through these experiments, we have successfully demonstrated the methyltransferase function of LcsG.

19) Page 6, lines 103-5: Once again, determining the source of the substrate is difficult. The authors must make it abundantly clear what the substrate was and where it came from, as well as that WT refers to the fungus. This may be clarified by referring to the following section.

Response: The substrate here was the ethyl acetate (EtOAc) extracts of the wild type (WT) of PLBJ-1 strain 14-days cultures. The “WT” refers to the wild type of *Purpureocillium lilacinum* strain PLBJ-1 (CGMCC3.17492). For detailed information on the extraction process, please refer to the “Culture Extraction” section within the Methods portion of the document. We have also modified this sections as your suggestions in page 7 lines 125-134.

20) Page 6, lines 112-14: How the precise substrate concentrations were adjusted in this range. Authors must elaborate in the text. Again, the authors can avoid confusion by adequately cross-referencing the methods section.

Response: For detailed methodology, including how these substrate concentrations were precisely adjusted, please refer to page 20 lines 436-440 “Michaelis-Menten enzyme kinetics” section in the methods part of the text.

21) Page 6, lines 114-16: This sentence appears to be missing text or has a typo, and thus needs to be corrected and rephrased.

Response: We have corrected and rephrase it as “The initial rate data were measured by LC-MS and fitted to the Michaelis-Menten equation to determine the kinetic parameters (Fig. 2D and Fig. S13). The K_{cat}/K_m values for LeuA (3) and LeuK0 (4) were found to be $65.39 \text{ s}^{-1}\text{M}^{-1}$ and $32.94 \text{ s}^{-1}\text{M}^{-1}$, respectively.”

22) Page 7, lines 130-40: This section should be rewritten to be more concise and readable.

Response: We have rewritten it. Please see this section page 8 lines 159-168.

23) Page 8, line 55: Change ...we employ... to ...we employed...

Response: We have changed “...we employ...” to “...we employed...”.

24) Page 9, lines 169-70: Add a bit more on mutating the five additional Asp residues (SAM binding) of LcsG to Ala in addition to the ones flanking and close to the N-terminus of LeuA to be methylated.

Response: We have incorporated the mutation details for mutating in the Methods section, specifically in "Gene cloning and plasmid construction" on page 16 lines 334-336. The sequencing results of the mutations are illustrated in Fig. S21A for reference (attached below).

```

      850      860      870      880      890      900
1 lcsG    CTCGAGGCCCTTGACTGGGCCGGAGCTGGCAAAGCCACCGTCGTTGACTTGGTGGCTCT
2 D296A   CTCGAGGCCCTTGACTGGGCCGGAGCTGGCAAAGCCACCGTCGTTGACTTGGTGGCTCT
3 D321A   CTCGAGGCCCTTGACTGGGCCGGAGCTGGCAAAGCCACCGTCGTTGACTTGGTGGCTCT
4 D348A   CTCGAGGCCCTTGACTGGGCCGGAGCTGGCAAAGCCACCGTCGTTGACTTGGTGGCTCT
5 K363A   CTCGAGGCCCTTGACTGGGCCGGAGCTGGCAAAGCCACCGTCGTTGACTTGGTGGCTCT
6 D368A   CTCGAGGCCCTTGACTGGGCCGGAGCTGGCAAAGCCACCGTCGTTGACTTGGTGGCTCT
7 D395A   CTCGAGGCCCTTGACTGGGCCGGAGCTGGCAAAGCCACCGTCGTTGACTTGGTGGCTCT

```

```

      910      920      930      940      950      960
1 lcsG    GCGGCTCACGACGACGTGCCCTTTGCCGAAAAATTCGCCGATCTCAAGATCAATCGTCCAG
2 D296A   GCGGCTCACGACGACGTGCCCTTTGCCGAAAAATTCGCCGATCTCAAGATCAATCGTCCAG
3 D321A   GCGGCTCACGACGACGTGCCCTTTGCCGAAAAATTCGCCGATCTCAAGATCAATCGTCCAG
4 D348A   GCGGCTCACGACGACGTGCCCTTTGCCGAAAAATTCGCCGATCTCAAGATCAATCGTCCAG
5 K363A   GCGGCTCACGACGACGTGCCCTTTGCCGAAAAATTCGCCGATCTCAAGATCAATCGTCCAG
6 D368A   GCGGCTCACGACGACGTGCCCTTTGCCGAAAAATTCGCCGATCTCAAGATCAATCGTCCAG
7 D395A   GCGGCTCACGACGACGTGCCCTTTGCCGAAAAATTCGCCGATCTCAAGATCAATCGTCCAG

```

```

      970      980      990      1000     1010     1020
1 lcsG    GACTGCTAGTTGCCAACCGAAAATTTGACGACGGCTACATCTCCGACGAGCTCAAGAAG
2 D296A   GACTGCTAGTTGCCAACCGAAAATTTGACGACGGCTACATCTCCGACGAGCTCAAGAAG
3 D321A   GACTGCTAGTTGCCAACCGAAAATTTGACGACGGCTACATCTCCGACGAGCTCAAGAAG
4 D348A   GACTGCTAGTTGCCAACCGAAAATTTGACGACGGCTACATCTCCGACGAGCTCAAGAAG
5 K363A   GACTGCTAGTTGCCAACCGAAAATTTGACGACGGCTACATCTCCGACGAGCTCAAGAAG
6 D368A   GACTGCTAGTTGCCAACCGAAAATTTGACGACGGCTACATCTCCGACGAGCTCAAGAAG
7 D395A   GACTGCTAGTTGCCAACCGAAAATTTGACGACGGCTACATCTCCGACGAGCTCAAGAAG

```

```

      1030     1040     1050     1060     1070     1080
1 lcsG    CGAGTGCTTCCTCGCTCATGACTTCTTACCCCTCAGCCCGTTCAGGCCGACATCTAC
2 D296A   CGAGTGCTTCCTCGCTCATGACTTCTTACCCCTCAGCCCGTTCAGGCCGACATCTAC
3 D321A   CGAGTGCTTCCTCGCTCATGACTTCTTACCCCTCAGCCCGTTCAGGCCGACATCTAC
4 D348A   CGAGTGCTTCCTCGCTCATGACTTCTTACCCCTCAGCCCGTTCAGGCCGACATCTAC
5 K363A   CGAGTGCTTCCTCGCTCATGACTTCTTACCCCTCAGCCCGTTCAGGCCGACATCTAC
6 D368A   CGAGTGCTTCCTCGCTCATGACTTCTTACCCCTCAGCCCGTTCAGGCCGACATCTAC
7 D395A   CGAGTGCTTCCTCGCTCATGACTTCTTACCCCTCAGCCCGTTCAGGCCGACATCTAC

```

```

      1090     1100     1110     1120     1130     1140
1 lcsG    CTCTTCAAAGTGGGCTTTTACGACTGGTCCAACAAGGACATCGTCAAGATCAATCAAGGCG
2 D296A   CTCTTCAAAGTGGGCTTTTACGACTGGTCCAACAAGGACATCGTCAAGATCAATCAAGGCG
3 D321A   CTCTTCAAAGTGGGCTTTTACGACTGGTCCAACAAGGACATCGTCAAGATCAATCAAGGCG
4 D348A   CTCTTCAAAGTGGGCTTTTACGACTGGTCCAACAAGGACATCGTCAAGATCAATCAAGGCG
5 K363A   CTCTTCAAAGTGGGCTTTTACGACTGGTCCAACAAGGACATCGTCAAGATCAATCAAGGCG
6 D368A   CTCTTCAAAGTGGGCTTTTACGACTGGTCCAACAAGGACATCGTCAAGATCAATCAAGGCG
7 D395A   CTCTTCAAAGTGGGCTTTTACGACTGGTCCAACAAGGACATCGTCAAGATCAATCAAGGCG

```

```

      1150     1160     1170     1180     1190     1200
1 lcsG    CTCGTGCTGCTCTCCGGCCGGGGGCTCGTGTCCCTCGTCTGGACTGATGGTGGACGTC
2 D296A   CTCGTGCTGCTCTCCGGCCGGGGGCTCGTGTCCCTCGTCTGGACTGATGGTGGACGTC
3 D321A   CTCGTGCTGCTCTCCGGCCGGGGGCTCGTGTCCCTCGTCTGGACTGATGGTGGACGTC
4 D348A   CTCGTGCTGCTCTCCGGCCGGGGGCTCGTGTCCCTCGTCTGGACTGATGGTGGACGTC
5 K363A   CTCGTGCTGCTCTCCGGCCGGGGGCTCGTGTCCCTCGTCTGGACTGATGGTGGACGTC
6 D368A   CTCGTGCTGCTCTCCGGCCGGGGGCTCGTGTCCCTCGTCTGGACTGATGGTGGACGTC
7 D395A   CTCGTGCTGCTCTCCGGCCGGGGGCTCGTGTCCCTCGTCTGGACTGATGGTGGACGTC

```

```

      1210     1220     1230     1240     1250     1260
1 lcsG    GGGCCCGAGGCTGCCCGGCTGATGCCCGGCTCGCTTCTGAAATACAGCAATGTGATTAGT
2 D296A   GGGCCCGAGGCTGCCCGGCTGATGCCCGGCTCGCTTCTGAAATACAGCAATGTGATTAGT
3 D321A   GGGCCCGAGGCTGCCCGGCTGATGCCCGGCTCGCTTCTGAAATACAGCAATGTGATTAGT
4 D348A   GGGCCCGAGGCTGCCCGGCTGATGCCCGGCTCGCTTCTGAAATACAGCAATGTGATTAGT
5 K363A   GGGCCCGAGGCTGCCCGGCTGATGCCCGGCTCGCTTCTGAAATACAGCAATGTGATTAGT
6 D368A   GGGCCCGAGGCTGCCCGGCTGATGCCCGGCTCGCTTCTGAAATACAGCAATGTGATTAGT
7 D395A   GGGCCCGAGGCTGCCCGGCTGATGCCCGGCTCGCTTCTGAAATACAGCAATGTGATTAGT

```

25) Page 9, lines 172-74: Rewrite the sentence to make it easier to read.

Response: We have revised it as “These findings are consistent with our earlier in vitro assays and molecular docking results. They indicate that the SAM binding site of LcsG is involved in leucinostatin methylation and provides additional support for LcsG being a SAM-dependent methyltransferase.”

26) Page 10, line 194: A separate section in the results section should be added to evaluate the biological activities of new leucinostatins.

Response: There is a separate section, please refer to the “Antifungal evaluation of leucinostatins” section in the Results part of the text.

27) Page 15, lines 309-11: How did these 10 mg of LeuA and K0 come about? How were other Leu substrates purified and quantified before use in assays? This should be clearly described in the methods section. The problem is that this information is not covered in the preceding sections, and there is no cross-referencing to the following sections.

Response: LeuA was obtained from a commercial supplier, while LeuK0 was extracted from the fermentation crude extract of the $\Delta lcsG$ strain. Additionally, LeuA0 was generated through the chemical methylation of LeuA, while LeuK2 and LeuK3 were synthesized by chemically methylating LeuK. We utilized the pure LeuA and LeuK0 for the methylation pilot study and employed the wild-type (WT) and $\Delta lcsG$ mutant crude extracts for bulk preparation, catering to the needs of subsequent experiments. These leucinostatins were dissolved in DMSO to prepare 10 mM stock solutions and subsequently diluted to the required concentrations for use in assays. To fix the problem about the information not covered in the preceding sections, we have changed the section “Chemical methylation of LeuA and LeuK0” to come before the previous section “Product purification of LeuK0, LeuA0, and LeuK3”.

28) Page 15, line 314: To make sense and avoid confusion regarding the use of purified substrate in the assays, this section must come before the previous section.

Response: We have moved the section “Chemical methylation of LeuA and LeuK0” before the previous section “Product purification of LeuK0, LeuA0, and LeuK3”.

29) Page 15, lines 315-18: These details, as well as the quantities of purified leucinostatins obtained and their purity levels, should be included in the supplementary file.

Response: We have added the information about the quantities of purified leucinostatins obtained and their purity levels into the section “Product purification of LeuK0, LeuA0, and LeuK3”.

30) Page 17, line 364: Change ...time cause assay... to ...time course assay...

Response: We have changed “...time cause assay...” to “...time course assay...”.

31) Page 17, lines 365-66: Change ...pH cause assay... to ...pH assay...

Response: We have changed “...pH cause assay...” to “...pH course assay...”.

32) Page 17, line 367: Change ...temperature cause assay... to ...temperature assay...

Response: We have changed “...temperature cause assay...” to “...temperature course assay...”.

33) Page 18, line 370: Change ...catalysts... to ...mutant catalysts...

Response: We have changed “...catalysts...” to “...mutant catalysts...”.

34) Page 18, line 380: This range is stated in the results as 0-200 μM . State the concentration ranges in a consistent and uniform manner.

Response: According to your suggestion, we have already added the concise concentration. Please see page 20 lines 436-439.

35) Page 19, line 406: What was the concentration of test compounds in the 10 μL solutions added to the plate wells?

Response: The concentration of the test compounds in the 10 μL solutions added to the plate wells was 25 μg per well. Please see page 22 line 465 and page 35 line 719.

36) Page 19, lines 409-10: Provide specifics about the ingredients of the rye agar that was used.

Response: The rye agar medium contained 50 g of crushed rye, 20 g of sucrose and 15 g of agar per liter.

37) Page 19, line 411: What was added to the wells that were aspirated is unclear. How the inhibition zone was determined?

Response: LeuK0, LeuA0, and LeuK3 were dissolved in DMSO at a concentration of 2.5 $\mu\text{g}/\mu\text{L}$. Subsequently, 10 μL of each compound was added to the respective wells, while 10 μL of DMSO was added to the control group well. The inhibition zone was observed for the clear zones and could be calculated by drawing circles of the zone. For determining the size of the inhibition zone, we can place the punched hole at the center of concentric circles, then observe which concentric circle the inhibition zone falls into. By examining the diameter scale of that particular circle, estimate the size of the inhibition zone.

38) Page 20, line 418: Add the strains that were used.

Response: We have changed “The strains were incubated at 25 $^{\circ}\text{C}$, and the MICs were determined at 48 hours for *C. neoformans* H99.” to “The *C. neoformans* H99 strains were incubated at 25 $^{\circ}\text{C}$, and the MICs were determined at 48 hours.”

39) Page 20, lines 412-22: This section needs to be rewritten so that the strains being tested are made crystal clear. It needs to be rephrased because it doesn't read well as it is.

Response: Thank you for this suggestion. We have rephrased this section as “To determine the minimal inhibitory concentration (MIC) following the recommendations of the National Committee for Clinical Laboratory Standards (NCCLS). We conducted the MIC assay with three replicates using the serial dilution method. We used 96-well plates with YM medium (containing 1% maltose extract and 0.2% yeast extract) for testing. Amphotericin B served as the positive control. In the experiment, the test compounds were first dissolved in DMSO and then serially

diluted in a growth medium. We monitored the results visually and also measured the optical density of the microplate wells, comparing them to positive and negative controls. The strains were incubated at 25 °C, and MIC values were determined after 48 hours for *C. neoformans* H99. To assess viability, we employed PrestoBlue resazurin dye (Life Technologies) as an indicator. The spectrophotometric MIC value was defined as the lowest concentration of a test compound that resulted in a culture with a density equivalent to 100% inhibition compared to the growth of the untreated control.”

40) Page 20, line 423: This needs to be expanded to include information on how the predicted structure was obtained using the Uni-fold. Here, more information about the step-by-step process is required.

Response: The sequence of LcsG has been submitted to the website <https://hermite.dp.tech>. JackHMMER with MGnify, JackHMMER with UniRef90, and HHBlits with Uniclust30 + BFD were used to search homology sequences to generate multiple sequence alignment (MSA). HHsearch was used to search structure templates from structures that were released before April 29th, 2020. MSA and the top 4 templates were combined as input and 100 models were generated.

41) Page 20, line 430: For repeatability, this section needs to be updated with enough details. Here, too, additional details regarding the methodical procedure are needed.

Response: The structure of leucinostatin A was modified from the structure of its analogy ZHAWOC6027(PDB: 8a19). The Diffdock webserver (<https://huggingface.co/spaces/simonduerr/diffdock>) was employed to simulate the dock. Upload the predicted LcsG dimer structure and leucinostatin A structure. Start the docking process and the website's backend algorithms will process the data to predict the binding poses of the ligand to the protein. Download the final docking poses and the first rank docking pose was selected as the model.

42) Page 31, line 602: For Fig. 4D, specify the assay, sample size, and statistics.

Response: In Fig. 4C, the sample size for this assay was 3 replicates used. Statistical analysis was performed using mean calculation. The update picture as below:

43) SI, Page 20: Give the sample size and the reason behind the absence of SD error bars in Fig. S13.

Response: In Figure S13, each point represents the mean of a set of parallel experiments (n=3). The absence of SD error bars was an oversight, and we appreciate your highlighting this issue.

We have rectified this by adding the appropriate SD error bars to Figure S13. A similar oversight was identified and corrected in Fig. 2C ,S3A, and S12. These adjustments enhance the clarity of our results, and the updated picture attached below.

Fig. 2C:

Fig. S3A:

Fig. S12:

Fig. S13:

Reviewer #2

1) While these results are certainly intriguing, the importance of these contributions is not clear, and I was not convinced by the mechanistic arguments. Furthermore, the text does not read wonderfully smoothly with many concepts referenced before they are defined, awkward transitions between topics, and a variety of grammatical/typographical errors.

Response: We have revised the manuscript for a smoother flow, addressed conceptual definitions, and corrected grammatical and typographical errors. Besides, we also try our best to improve the presentation of mechanistic arguments. We analyzed the relationship between D368 and D395 by MD simulations, please see Fig 4B and pages 203-205.

2) With regards to impact, the authors state that “NMT-catalyzed iterative N-methylation at the terminus of NRPs has been rarely reported.” However, there is no reference or context provided here. Has it been reported previously, or not, to the best of the authors’ knowledge? Without this clarification, the novelty of the described reactions is difficult to assess.

Response: After conducting an extensive literature review, we have identified only one published article that reports NMT-catalyzed N-methylation at the terminus of NRP part within a NRP-PK hybrid compound, as mentioned in our introduction section. Apart from this specific instance, we have not come across any other articles discussing N-methylation of NRP termini in the existing literature.

3) Likewise, on page 7, the authors reference leucinostatins as well-known antibiotics, but there are no references here. They then proceed to discuss their antifungal assays. Why perform antifungal assays if these molecules are commonly antibiotics? Are the molecules in this study also antibiotics? Are their antifungal properties unusual for the family of NPs? Why were the particular strains selected for testing in the first place? How do the MICs compare to other

relevant molecules? Clarification is needed here to place the presented results in context with the broader scientific literature in this area.

Response: In this research, we selected the human pathogen *Cryptococcus neoformans* H99 and the plant pathogen *Phytophthora infestans* as the test strains. The reason for choosing *Cryptococcus neoformans* H99 is that in the original paper where LeuA and LeuB were first isolated (Fukushima K, Arai T, Mori Y, et al. Studies on peptide antibiotics, leucinostatins I. separation, physico-chemical properties and biological activities of leucinostatins A and B [J]. The Journal of Antibiotics, 1983, 36(12): 1606-1612.), the authors used *Cryptococcus neoformans* as the test strain to determine the MIC value of them. *Cryptococcus neoformans* strain H99 was the most widely used *C. neoformans* reference strain globally. It is a highly virulent and drug-resistant strain. Therefore, we wanted to investigate the activity of leucinostatins against the H99 strain.

In the work of our research group led by Gang Wang (Wang G, Liu Z, Lin R, et al. Biosynthesis of antibiotic leucinostatins in bio-control fungus *Purpureocillium lilacinum* and their inhibition on *Phytophthora* revealed by genome mining [J]. PLoS pathogens, 2016, 12(7): e1005685.), leucinostatins were studied for their inhibitory activity against *Phytophthora infestans*. Hence, we chose this strain for our activity experiments.

4) The least compelling aspect of the manuscript was the mechanistic hypothesis, which primarily relies on the AI-predicted structure and subsequent docking simulations. As the structure is predicted, not experimental, it is important to address the confidence of the model. It is noted in the Methods section that the average pLDDT score was 0.88. While this is certainly reassuring, it does not provide insight into what parts of the structure were lower confidence. If active site loops/residues are lower confidence, then the confidence in the docking models is equally questionable. A discussion of the pLDDT surrounding the active site is warranted. A figure of the overall structure colored by pLDDT may also be useful depending on the distribution of uncertainty. Along these same lines, were any residues considered flexible in the docking simulations? If the AI model was less confident about particular residues/sidechains, these should not be static in the docking simulation.

Response: We have incorporated a figure (refer to Fig.S18A, attached below) depicting the comprehensive structure with color-coded pLDDT scores. This visualization offers a detailed representation of the structural confidence, where a blue hue corresponds to a pLDDT score of 1.0, indicating high confidence, while a red hue signifies a pLDDT score of 0, representing lower confidence. This color scheme enhances the clarity of the structural reliability assessment in our AI-predicted protein model. Additionally, we included Fig.S18B (attached below), highlighting the pLDDT scores of the six predicted active site residues. The corresponding pLDDT scores for the six predicted active sites are now available in Table S3 for further clarity.

5) Why was only the top ranked docking pose considered? How different was this pose to the next highest ranked pose? What was the difference in the ranking? These rankings are far from perfect. Did the authors consider any single point energy calculations or short MD simulations to assess stability of the poses to ensure an optimally modeled interaction. This seems especially relevant given attempts to infer mechanistic behavior. A figure depicting the distances between the putative catalytic residues, the targeted position, and the SAM methyl in the proposed model would help to motivate these discussions.

Response: We acknowledge the challenges posed by the size of leucinostatins and the difficulties encountered in obtaining pose results through docking and induced fit docking using various leucinostatin structures (LeuA, LeuB, LeuK0, LeuK1). Despite our extensive efforts, we were unable to obtain conclusive results from these initial docking attempts. In response to these challenges, we turned to the results obtained from diffdock for further investigation through MD simulations. The MD results revealed a stable interaction between N and D368, providing valuable insights into the potential binding mechanisms. This observed interaction aligns well with the outcomes of our multiple sequence alignment and protein mutagenesis experiments. Specifically, the consistent results obtained from these complementary approaches lend support to our hypothesis suggesting a coordinated action between residues D368 and D395 in the methylation mechanism of *N*.

6) Furthermore, while mutagenesis studies highlight the importance of D368 and D395 in catalysis, these results do not provide an airtight case for their involvement in the deprotonation step. Are there other residues nearby that could also potentially serve in this capacity? Is it possible one or both residues are solely important for binding? How many other interactions hold the substrate in place? Does a double mutant abolish activity?

Response: Thank you for your valuable suggestions. Unfortunately, despite numerous attempts for purifying the D368 and D395 double mutant protein, including changing the *E. coli* strains, we were unable to achieve sufficient protein purity. The sequencing results of the double mutant vector are shown in Figure A; the small-scale purification of the double mutant transformant protein is presented in Figure B; with two representative results from multiple protein purifications shown in Figure C. As evident from the data, the solubility of the protein decreases upon the double mutation, with the majority of the protein found in the insoluble fraction. We acknowledge the importance of exploring alternative residues in proximity to D368 and D395 for potential involvement in catalysis or binding. However, obtaining the double mutant protein in sufficient purity poses a challenge for us. Moreover, the poor solubility of the protein after the

double mutation gives the hint that these two residues may be important for the structural and stability of LcsG, hindering us from conducting the *in vitro* enzymatic assay.

7) The authors state that “the deletion of *lcsG* only led to the abolishment of LeuB and LeuA, suggesting...” However, there still appears to be a small peak in Figure 1c around the elution time of LeuA in the trace for the *LcsG* knockout. Is this something else? This should be clarified.

Response: The PLBJ-1 strain contains numerous compounds within the leucinostatin family, in addition to the four major metabolites, LeuC, LeuA, LeuB, and LeuK0. The m/z $[M+H]^+$ of the small peak observed in Fig. 1C is 1216.8. This compound was present in the strain before the knockout. Through HRESI-MS-MS analysis, we have confirmed that this compound is not LeuA. Additionally, this unknown compound shares the same C-terminus as LeuK0 (as shown below), which may explain why it persists after the knockout of *lcsG*. We add the indication of this unknown compound in Fig. 1C, along with its ESI-MS-MS information in the Table S1.

Fig. 1C:

Table S1:

	B1	A2	B2	B3	B4	B5	B6	B7	B8	B9	C10	Y5	Y4	Y3	Y2
LeuA	111	194	222	435	564	649	762	875	960	1045	1173	457	344	259	174
LeuB	111	194	222	435	564	649	762	875	960	1045	1173	443	330	245	160
LeuC	111	194	222	435	564	649	762	875	960	1045	1173				
LeuA0	111	194	222	435	564	649	762	875	960	1045	1173				
LeuK0	111	194	222	435	564	649	762	875	960	1045	1173	473	360	275	190
Unknown compound	111	194	222	417	546	613	744	857	942	1027	1155	473	360	275	190
LeuK1	111	194	222	435	564	649	762	875	960	1045	1173	675	562	477	392
LeuK2	111	194	222	435	564	649	762	875	960	1045	1173	487	374	289	204
LeuK3	111	194	222	435	564	649	762	875	960	1045	1173				

8) The experiments to assess kinetics on page 6 were very confusing. The authors state that they used “the crude extract of WT as the substrate”? The accumulation of various products is then described without a clear description of the experiments. The Fig. 2c caption is equally unhelpful. Was only 2 or 4 provided at the start of the reaction, or were both 2 and 3, or 4 and 6 provided for these assays? How can the peak areas for what I presume are the selected substrates for each reaction be negative? While I can see the ‘slight increase’ the authors note for 3, I do not see the same for 6 in the Fig. 2c data.

Response: “the crude extract of WT” refers to “the ethyl acetate (EtOAc) extracts of the WT of PLBJ-1 strain cultures”. This extract inherently contains **2**, **3**, and **4**, meaning these components are present from the beginning of the reaction. In Fig. 2C, we depict the differences in component levels between the experimental and control groups. Positive values indicate that a component's level in the experimental group is higher than in the control group, suggesting accumulation during the catalysis. Conversely, negative values suggest that the component's level in the experimental group is lower than in the control group, indicating consumption during the catalysis. The negative values for **2** and **4** can be attributed to their depletion as substrates at the beginning of the catalytic reaction. On the other hand, the positive values for **3** and **6** indicate their accumulation at the start of the catalytic reaction, with the decreasing differences over time suggesting their subsequent consumption. We have revised the context and the picture to make it clarify. Please see Fig. 2C (attached below) and page 7 lines 126-134.

9) Whenever structural alignments are discussed/depicted, the authors should not the RMSDs. Are these alignments of just the backbone? How many atoms were included, etc.? These values are important for gauging similarity.

Response: We have created the structural alignments of LcsG and its homologies, please see Fig. S17B (attached below). Additionally, we showed the RMSDs at Table S2, please see it. All these alignment and RMSD calculations were performed using the alpha carbons of amino acids.

10) The author's state that accurate stereochemistry of leucinostatins is hard to predict before basing their structure of LeuA off the structures of similar molecules. How confident are they in this stereochemical assignment? Did they try docking alternatives?

Response: Facing challenges in obtaining results from docking and induced fit docking with various leucinostatin structures, we opted to optimize their stereostructures. As mentioned in our manuscript, the leucinostatins, coupled with non-standard amino acid residues, rendered accurate prediction challenging. In response, we referenced crystal structures of two reported analogs for optimization, as illustrated in Fig. S19 (attached below). The structures of these analogs exhibited partial similarities to LeuA, and combining their features provided a basis for generating a plausible stereostructure for LeuA.

We have made diligent attempts to dock various leucinostatins (LeuA, LeuB, LeuK0, LeuK1). However, none yielded informative poses.

LeuA

ZHAWOC6027

helioferin A

11) Did the authors consider generating a SSN as well a phylogenetic tree? This approach may provide additional insights into the relationship between these enzymes or help to identify other similar systems.

Response: In response to the reviewer's comment, we have indeed incorporated a SSN analysis instead of the phylogenetic tree, as reflected in Figure S22 (attached below). This SSN, derived from 5000 sequences, offers a more comprehensive overview of the relationship between LcsG and its homologous sequences compared to the initial phylogenetic tree. We appreciate the suggestion and believe that the inclusion of the SSN strengthens the robustness of our study.

12) In the discussion, the authors compare the catalytic residues of OxaC and CHOMT to LcsG. This comes out of nowhere. Why reference these two specific enzymes? While Fig. S22 purportedly corresponds to this discussion, I don't see these two enzymes in that sequence alignment...?

Response: The comparison of the catalytic residues of LcsG with OxaC and CHOMT was made based on structural analyses conducted using the SWISS-MODEL. Our search for homologous structures in the SWISS-MODEL repository revealed that both OxaC and CHOMT contain either SAM or SAH within their structures, making them suitable for direct structural comparison with LcsG.

We would like to clarify that OxaC and CHOMT are included in Fig. S17A (attached below) of the Supplementary Information. The enzyme OxaC is referenced as A0A1B2TT09, and CHOMT is referred to as P93324. Now we revised to clarify it in the caption. We apologize for any confusion caused and appreciate the opportunity to clarify this point in our manuscript.

```

      260      270      280      290      300      310
1 LcsG  KGWRQRDMVESLRLAKEKMGAESALLEALDWA.GAGKATVVDLGGSGHDDVPLAEKFP
2 LepI  TPNFLSTF.....PLEKELGSWSAEPEKALFVDLGGGMCHACIRLRKYP
3 ChOMT NQIFNKSMDVDCATE...MK...RM...LEIYTG...FEGISTLVDGGGSRNLELTIISKY
4 OxaC  ADVQCIAMDL.....Y...PW...ERLSDA.KGSNATLVDLGGSGHNGTRAIWALAP
5 RedM  LDAFQQAMTG.....LSMRSAAHAL.AEAIDWS.AY...RTVADLGGCAECVLIHLLELRRHP
6 PhzM  GRRFLAMKA.....SNLAFH.EI...PRLLDFR...G...RSEVDVGGGSGELTKAILQOEP
7 MmcR  RELFNRAAGS.....VSLTEAGQV.AAAYDFS.GA..ATAVDLGGGRGSLMAAVLDAFP
8 Rosa  RDEFDAAAVE.....FGQYFADDE.LTSFDFEG.RF...TRFADLGGGRGQFLAGVLTAVP

```

```

      320      330      340      350      360
1 LcsG  D...KIIVQDLPSQCPKFDGGYISDELKRVVSLAHDFFT..PQPVAADIVLFRWVFD
2 LepI  NQPG.RVILQDLPPVLAQAQATLPLSGIEM...PHNFHT.PQPVQAKFYFLRLILRD
3 ChOMT LI...KGINFDLPQVIENAP...PLSGIEH...VGGDMFA...SVFQGDAMILKAVCHN
4 OxaC  KLNGCRFIVQDLPEVIGHSQALRAEGIEPO...VYDFLKQEQPVHGASIVYFRRVFHD
5 RedM  HL...RGTGFDLAAVRPSPQRRHEESGLGDRLAFRAGDFFA..EPLQADALVFGHILSN
6 PhzM  SA...RQVMDLREGSLGVARDNLSLLAGERVSLVGGDMLQ..EVPNNGDIVYLLSRITGD
7 MmcR  GL...RGTLLRPPVAEEARELLTGRGLADRCEILPGDFE..TIPDGAIVYLLKRVLHD
8 Rosa  SS...LGVLVGPAVVASAHKFLASQNLTERVEVRIGLDFD..VLPVGCDAIVLGGVLD

```

```

      370      380      390      400      410      420
1 LcsG  WSNKD.IVKTIKALVPAAL..RPGAQVVLVLDLMDVVG.PEAAAVMPRSLLYSNVISLKT
2 LepI  FPDHQ.ALEILQNIIVPAM...DAESRIVIVDGVVPEK...GARW...AETGTDICI
3 ChOMT WSDKE.CHEFLSNCHKAL..SPNGKVIIVFELPPE.PNTSEE...SKLVSTLDNLMF
4 OxaC  WFDLPECKKILDNTRAM.SREHSRLHDIIVPEI...GATM...SHAWODLSL
5 RedM  WALPK.AKTLLLRKAHEAL..PEGGIVVIVETLIDERRRNVVGL.....LMSLTML
6 PhzM  LDEAA.SIRLLIGNCREAM..AGDGRVVVIERLISASEPS.PMSV.....LWDVHLF
7 MmcR  WDDDD.VVIRLRIATAM..KPDSTRLLVIDNLDIER.PA.ASTL.....FVDLLLL
8 Rosa  WADAD.AVRLVRIROAMGDAPEARLLLDVIGET.GE.LG.K.....VLDLDM

```

13) Figure 1, panel c – What is the top smaller panel here? Is it a zoomed-in view of the first peak? While the position of the peak on the x-axis appears to suggest it is not (i.e. the peaks in the top spectrum tail off around 15.6 min, while those in the bottom panel tail off around 16 min), this is very confusing.

Response: The top smaller panel in Fig. 1C, is indeed a zoomed-in view of compound 1. Compound 1 is located to the left of compound 4 in the bottom panel. Its signal peak is much smaller and almost connected to compound 4, making it difficult to distinguish. The peaks that tail off around 16 min in the bottom panel you mentioned correspond to compound 4 instead of compound 1. Now we revised it as below, please see it.

14) Compounds 6-8 are referenced in the text and in Fig. 2 before they are even introduced. They are not described until the reader gets to Fig. 3. To avoid confusion, a figure of these compounds needs to be presented earlier in figure or text.

Response: Thanks for your suggestion, we now presented them in Fig. 1A (attached below) to avoid confusion, please see it.

15) Currently, there are no figures of the full AI prediction of the overall enzyme structure. Such an image would allow the authors to label the different domains referenced in the text. Furthermore, a comparison with other related NMTs and OMTs would be very helpful to provide a visualization of the overall similarities or lack thereof.

Response: We have generated a comprehensive figure illustrating the full AI prediction of the overall enzyme structure of LcsG. This new figure provides a detailed representation of different domains referenced in the text. Additionally, to enhance clarity and facilitate comparison, we have included images showcasing the structural alignment of LcsG with other related NMTs and OMTs. Please see Fig. S18.

16) Comparison between OMT and NMT mechanisms was exceedingly difficult without schemes for reference. The sequence alignments of Fig. S17, S19 and S22 were also poorly captioned, difficult to correlate with the discussion in the text.

Response: In response to the concern raised, we have revised the relevant section to facilitate a clearer understanding of the OMT mechanisms. Additionally, we have improved the captions of supplement figures to ensure better correlation with the corresponding discussion in the text. Please refer to page 13 line 262-274.

17) Generally speaking, the resolution of figures in the SI is problematic. This is particularly true for S9 and S14, which are difficult to read both due to their size and resolution.

Response: We have reformatted the images in the SI to high resolution and reorganized the layout for enhanced readability. Please see it.

Fig. S9:

Fig. S14:

Fig. S15:

18) The sequence alignment in Fig. S17 contains a series of boxed, highlighted, marked residues. However, there is no indication in the caption as to what the red arrows, red highlighting and blue boxes are pointing out exactly. Either a lengthier caption, or legend is warranted here. It would be helpful to label the SAH/SAM binding motif, proposed catalytic residues, etc. This figure should also be referenced in the text when discussing these features.

Response: We have revised the caption of Fig. S17 to include a more detailed legend. We have also included labels for important features such as the SAH/SAM binding motif and proposed catalytic residues, please see Fig. S17.

19) There is confusing text throughout. A couple of selected examples are listed below, but these are scattered throughout contributing to confusion. Perhaps it would be worth investigating an editing service. Note that the plural of C-terminus is C-termini not C-terminuses.

Response: We have changed all of “C-terminuses” to “C-termini”

20) - “However, the biosynthetic mechanisms of the diverse C-terminuses remain unknown.” I presume the authors meant the biosynthetic mechanisms to generate the diverse C-termini as the C-termini don’t have mechanisms?

Response: We have changed “However, the biosynthetic mechanisms of the diverse C-terminuses remain unknown” to “However, the biosynthetic mechanisms to generate the diverse C-termini remain unknown”.

21) - The authors write “In this work, we identify a discrete NMT that could catalyze a unique moiety located in the terminus of a NRP, named as LcsG, from the biological control fungus *P. lilacinum* PLBJ-1.23”. “catalyze a unique moiety” doesn’t make sense. As is, the text implies LcsG is the NRP, but it is the modifying enzyme. This would read more smoothly as follows: “In this work, we identify a discrete NMT, LcsG, that catalyzes the installation of a unique moiety located at the terminus of an NRP from the biological control fungus *P. lilacinum* PLBJ-1.23”

Response: We have changed “In this work, we identify a discrete NMT that could catalyze a unique moiety located in the terminus of a NRP, named as LcsG, from the biological control fungus *P. lilacinum* PLBJ-1.” to “In this work, we identify a discrete NMT, LcsG, that catalyzes the installation of a unique moiety located at the terminus of an NRP from the biological control fungus *P. lilacinum* PLBJ-1.”

Reviewers' comments:

Reviewer #1 (Remarks to the Author):

Overall, the paper has significantly improved from its previous version, addressing missing information and reducing confusion. However, there are still some issues that need attention before it can be considered for publication.

Major comments:

Throughout the manuscript, there is inconsistency in the use of abbreviations. For example, in figures, there is inconsistency in the use of amino acid abbreviations (single and triple letter codes). Standardizing the usage of amino acid abbreviations across figures would enhance clarity and consistency. Unless the residues depicted are indicating a mutational transition, they should be in triple letter code keeping it consistent with the text.

Some sections of the paper appear disjointed or appended, rather than seamlessly woven into the narrative flow. Revision of section transitions, as well as ensuring that each section contributes cohesively to the study's overall story, would improve readability and comprehension. The abstract, as well as a few sections of the manuscript, could benefit from improved flow and readability. Careful editing to ensure smooth transitions between ideas, as well as better information organization, would improve the paper's overall consistency.

To remove typos and grammatical errors from the text, the language must be meticulously edited.

Minor comments:

Page 2: The authors have made several changes to the abstract since the previous version. The current version, in my opinion, needs some changes to improve its readability.

Page 2, lines 2-3: Consider revising the title with a minor change to: "Change to - Characterization of a methyltransferase for iterative N-methylations at the leucinostatin termini in *Purpureocillium lilacinum*".

Page 2, lines 19-20: Change to: ...N-methyltransferase (NMT)-catalyzed methylations at nonribosomal peptide termini have rarely been reported...

Page 2, line 20: Change ...N-methyl... to ...N-methylation...

Page 2, lines 21-22: Change ...essential to the methylation of leucinostatins... to ...essential for leucinostatin methylation.

Page 2, line 22: Change ...In vitro assay by and HRESI-MS-MS analysis proved... to ...In vitro assays and HRESI-MS-MS analyses revealed...

Page 2, line 24: Change ...which showed effective... to ...which demonstrated effective...

Page 2, line 28: Change ...Diffdock and the molecular dynamic simulation... to ...Docking and the molecular dynamic simulation studies...

Page 2, lines 29-31: Change ...These findings provide an approach for enriching the variety of natural bioactivity of NRPs and give a hint of the potential catalytic mechanism underlying N-methylation of NRPs... to ...These findings suggest a method for increasing the variety of natural bioactivity of NRPs and a possible catalytic mechanism underlying N-methylation of NRPs.

Page 3, line 41: Change... the nature of the N-methyl amino acid... to ... their nature of the N-methyl amino acids...

Page 3, lines 47-51: Authors should consider rephrasing this for greater clarity.

Page 3, lines 50-53: Consider changing this sentence to: ...However, there are few reports on the terminal N-methylation of NRPs. The literature reviewed primarily focuses on single-site N-methylation within NRP structures...

Page 4, lines 56-57: Consider changing this sentence to: ...To our knowledge, the terminus of NRPs has never before been shown to undergo a discrete NMT-catalyzed iterative N-methylation...

Page 4, line 70: Change ...catalyzes the installation of... to ...catalyzes the incorporation of...

Page 4, lines 71-2: Authors should provide full forms of abbreviations and shortforms when they first appear in the manuscript's main text, for example. *P. lilacinum* strain PLBJ-1 should be written as *Purpureocillium lilacinum* strain PLBJ-1.

Page 4, lines 73-5: Change ...structure elucidations of products demonstrated the LcsG conducts iterative methylation at the terminal-free amines of leucinostatins... to ...structure elucidation of the products demonstrated that LcsG is involved in the iterative methylation of terminal-free amines in leucinostatins...

Page 8, line 150: Change ...with those data of LeuA (3)... to ...with those of LeuA (3)...

Page 9, line 171: The authors should specify the purification technique so that readers can understand without having to refer to the methods section.

Page 9, line 185: The author state here: ...employed artificial intelligence (AI) methods to approximate a model of LcsG...The authors should specify which online AI predictive tool they used, such as Uni-Fold, and include a link to it in the methods section.

Page 10, line 194: When acronyms like SAH appear in the manuscript text for the first time, define them in their entirety.

Page 10, lines 198 & 205: When using amino acid residue abbreviations in model structures, the authors should be consistent. In the text, Figures 4A–B, and S18, they seem to use single- and triple-letter abbreviations interchangeably.

Page 10, line 209: Change ...showed decreases in the conversion of LeuA (3)... to ...showed a decrease in the conversion of LeuA (3)...

Page 10, lines 210-11: Change ...diffdock and MD simulations results... to ... docking and MD simulations results...

Page 11, line 230: Change ...the LcsG-D368A and LcsG-D395A showed obvious decreases... to ...the mutants D368A and D395A showed obvious decreased methylation...

Page 12, line 241: Change ...capacity of catalyzing catalyzes two methylations... to ...capacity for catalyzing two methylations...

Page 12, line 213: Change ...followed by released from... to ...followed by release from...

Page 15, line 306: Change ...filtration cultivation for... to ...filtration after cultivation for...

Page 16, line 345: Change ...with 400 µg/mL G418 (Inalco, USA) (400 µg/mL) for 3-5 days... to ...with 400 µg/mL G418 (Inalco, USA) for 3-5 days...

Page 18, line 391: Change ...dimethylformamide (THF)... to ...dimethylformamide (DMF)...

Page 23, line 485: Change Uni-fold to Uni-Fold and include the URL.

Page 23, line 489: Change mafft and esript3 to MAFFT and ESPript 3 respectively, and include their URLs.

Page 23, lines 494-98: Consider rephrasing to: ...The predicted LcsG dimer structure, and leucinostatin A structure was uploaded to initiate the docking process, and the website's backend algorithms processed the data to predict the ligand's binding poses to the protein. The model was then chosen from the final docking poses as well as the top-ranked docking pose...

Page 24, line 510: Change ...E-Value threshold... to ...E-value threshold...

Reviewer #2 (Remarks to the Author):

The manuscript is much improved following clarifications to the text and the additional, primarily SI, figures. Expansion of the introduction provides much needed contextualization and emphasizes the significance of the work. Likewise, the added detail in the discussion of enzyme activities, among other experimental details, was very helpful.

I agree with the authors that the addition of sequence similarity analysis enhanced the manuscript. However, in the current form, evidence the support their hypothesis that NMTs are disguised as OMTs by simple sequence analysis is somewhat limited. The authors' arguments could be more compelling given some quick additional analysis. Firstly, the authors may be able to distinguish between OMTs and NMTs more easily by performing a new SSN with JUST the sequences from the orange cluster. Additionally, the authors discuss at length key differences in active site residues that distinguish OMTs from NMTs. It would be much more convincing if the

authors manually curated their analysis to characterize sub-clusters within the orange cluster by these residues. The inclusion of sequence logos may also aid in the visualization of this analysis.

Do the authors think this is a case of convergent evolution? This question may be answered by a phylogenetic analysis.

While the above suggestions require additional analysis, they are not incredibly time consuming such that they would delay publication substantially. They are, however, crucial to supporting the authors' ultimate conclusions.

Additional minor questions/critiques are detailed below.

The authors provided an explanation of their antifungal assays along with why certain strains were selected in their point-by-point response but did not alter the text itself. Without any clarification, this remains confusing for the reader. By no means am I suggesting that the authors incorporate their lengthy response, but minor changes could help clarify this section. E.g.

“Leucinostatins are well-known [antimicrobial and antimycotic compounds]. We carried out the purification for the compounds and obtained LeuA0 (8), LeuK0 (4) and LeuK3 (7) in sufficient quantities to do the antifungal assay. We determined the inhibitory activity of these compounds against the drug resistant strain *C. neoformans* H99 and the plant pathogen *P. infestans*, [both of which have been shown to be inhibited by LeuA and LeuB,(references)] by agar diffusion assays.”

As DiffDock is a very niche software, the term ‘molecular docking’ seems more appropriate for use in the abstract. E.g. “Molecular docking and dynamics simulations showed...”

Likewise, the second ‘D’ in ‘DiffDock’ is also capitalized.

The text still does not read wonderfully smoothly and there are a variety of grammatical and typographical errors. Perhaps it would be worth investigating an editing service. Two examples that are immediately apparent in the introduction are below, but there are others throughout.

“One study that discusses the N-methylation of the pipercolic acid moiety, which serves as the initial substrate for tubulysin synthesis.” This is not a complete sentence, so it is unclear what the authors are trying to convey.

“To our knowledge, a discrete NMT-catalyzed iterative N-methylation is not previously observed in the terminus of NRPs.” The verb tenses are mixed here, along with a missing pluralization. Something like “To our knowledge, a discrete NMT-catalyzed iterative N-methylation has not previously been observed in the termini of NRPs.” Would be clearer.

The links in the SI table of contents are not all associated with the correct sections in the document.

The additional figures depicting the complete structure of the enzyme are certainly worthwhile, but I nonetheless encourage the authors to label the termini as the roles of both are described in the text.

Upon discussing the pLDDDT scores of the overall structure as well as specific residues on page 9, the authors could make their argument more easily digestible by clearly stating that AI predicts the structure with high confidence, including active site residues proposed to be relevant for substrate binding and catalysis.

Did the authors mean 'subsequent molecular dynamics' when they said "The following molecular dynamics (MD) simulations"? 'Following' implies they will talk about those simulations immediately after that sentence, but that is not the case.

On line 223, the authors use mutant data as support for their argument, but this is before the mutant data have been discussed (primarily lines 230-231). If the authors simply remove reference to the mutant data in line 223, the paragraph will likely read more smoothly. The introduction of that data in lines 230-231 follows nicely from the computational discussion.

The authors describe 'boxes' in their SSN. Is this outdated? Their current SSN seems to be organized by color, not boxes.

Characterization of a methyltransferase for iterative *N*-methylations at the leucinostatin termini in *Purpureocillium lilacinum*

Reviewer #1:

Major comments:

- 1) Throughout the manuscript, there is inconsistency in the use of abbreviations. For example, in figures, there is inconsistency in the use of amino acid abbreviations (single and triple letter codes). Standardizing the usage of amino acid abbreviations across figures would enhance clarity and consistency. Unless the residues depicted are indicating a mutational transition, they should be in triple letter code keeping it consistent with the text.

Response: Thank you for pointing out the inconsistencies in the use of abbreviations. We have revised the manuscript and standardized all amino acid abbreviations to single-letter codes as suggested. Please refer to Figure. S18C; line 28 on page 2, line 187 on page 9, lines 209 and 213 on page 10, lines 214-216 and 235 on page 11, line 237 on page 12, line 345 on page 16, for the updated abbreviations.

Figure. S18C:

- 2) Some sections of the paper appear disjointed or appended, rather than seamlessly woven into the narrative flow. Revision of section transitions, as well as ensuring that each section contributes cohesively to the study's overall story, would improve readability and comprehension. The abstract, as well as a few sections of the manuscript, could benefit from improved flow and readability. Careful editing to ensure smooth transitions between ideas, as well as better information organization, would improve the paper's overall consistency.

Response: We acknowledge the issues you pointed out regarding the disjointed sections and we have undertaken a comprehensive revision to enhance the transitions and overall flow. We have improved the abstract and other noted sections to ensure that they contribute effectively to the manuscript's narrative. Additionally, we have refined the organization of information throughout

the paper to guarantee a smoother transition between ideas, aiming to improve both readability and the paper's overall consistency.

- 3) To remove typos and grammatical errors from the text, the language must be meticulously edited.

Response: We have addressed this concern by engaging a professional editing service to thoroughly review and polish the language in our manuscript. Please refer to the yellow-highlighted sections in the manuscript. We believe this has significantly enhanced the clarity and readability of the text.

Minor comments:

- 1) Page 2: The authors have made several changes to the abstract since the previous version. The current version, in my opinion, needs some changes to improve its readability.

Response: Thanks for your suggestion. We have had an editing service revise the sections you mentioned to enhance readability. Please refer to the yellow-highlight sections.

- 2) Page 2, lines 2-3: Consider revising the title with a minor change to: “Change to - Characterization of a methyltransferase for iterative *N*-methylations at the leucinostatin termini in *Purpureocillium lilacinum*”.

Response: Incorporating feedback from your comment and the editing service, the title has been changed to “Characterization of a methyltransferase for iterative *N*-methylation at the leucinostatin termini in *Purpureocillium lilacinum*”. Please see Page 1, lines 1-2.

- 3) Page 2, lines 19-20: Change to: ...*N*-methyltransferase (NMT)-catalyzed methylations at nonribosomal peptide termini have rarely been reported...

Response: Incorporating feedback from your comment and the editing service, this part has been changed to “*N*-methyltransferase (NMT)-catalyzed methylation at the termini of nonribosomal peptides (NRPs) has rarely been reported.” Please refer to page 2, lines 19-20 for verification.

- 4) Page 2, line 20: Change ...*N*-methyl... to ...*N*-methylation...

Response: Modifications have been made in accordance with your suggestion. Please refer to page 2, line 21 for verification.

- 5) Page 2, lines 21-22: Change ...essential to the methylation of leucinostatins... to ...essential for leucinostatin methylation.

Response: Modifications have been made in accordance with your suggestion. Please refer to page 2, line 22 for verification.

6) Page 2, line 22: Change ...In vitro assay by and HRESI-MSMS analysis proved... to ...In vitro assays and HRESI-MSMS analyses revealed...

Response: Modifications have been made in accordance with your suggestion. Please refer to page 2, line 22 for verification.

7) Page 2, line 24: Change ...which showed effective... to ...which demonstrated effective...

Response: Modifications have been made in accordance with your suggestion. Please refer to page 2, line 24 for verification.

8) Page 2, line 28: Change ...Diffdock and the molecular dynamic simulation... to ...Docking and the molecular dynamic simulation studies

Response: Modifications have been made in accordance with your suggestion. Please refer to page 2, line 24 for verification.

9) Page 2, lines 29-31: Change ...These findings provide an approach for enriching the variety of natural bioactivity of NPRs and give a hint of the potential catalytic mechanism underlying N-methylation of NRPs... to ...These findings suggest a method for increasing the variety of natural bioactivity of NPRs and a possible catalytic mechanism underlying N-methylation of NRPs.

Response: Incorporating feedback from your comment and the editing service, this part has been changed to “Thus, this study suggests a method for increasing the variety of natural bioactivity of NPRs and a possible catalytic mechanism underlying the *N*-methylation of NRPs.” Please refer to page 2, lines 29-31 for verification.

10) Page 3, line 41: Change... the nature of the N-methyl amino acid... to ... their nature of the N-methyl amino acids...

Response: Modifications have been made in accordance with your suggestion. Please refer to page 3, line 41 for verification.

11) Page 3, lines 47-51: Authors should consider rephrasing this for greater clarity.

Response: Modifications have been made in accordance with your suggestion: “For instance, in the production of bioactive pentapeptides known as cycloaspeptides, the NRPS lacks NMT domains. Instead, an independent NMT is partnered with the NRPS to supply methylated substrates, preferentially incorporating methylated amino acids at two specific positions within the cycloaspeptides”. Please refer to page 2, lines 47-50 for verification.

12) Page 3, lines 50-53: Consider changing this sentence to: ...However, there are few reports on the terminal *N*-methylation of NRPs. The literature reviewed primarily focuses on singlesite *N*-methylation within NRP structures...

Response: Modifications have been made in accordance with your suggestion. Please refer to page 3, lines 50-53 for verification.

13) Page 4, lines 56-57: Consider changing this sentence to: ...To our knowledge, the terminus of NRPs has never before been shown to undergo a discrete NMT-catalyzed iterative N-methylation...

Response: Modifications have been made in accordance with your suggestion. Please refer to page 4, lines 54-56 for verification.

14) Page 4, line 70: Change ...catalyzes the installation of... to ...catalyzes the incorporation of...

Response: Modifications have been made in accordance with your suggestion. Please refer to page 4, line 71 for verification.

15) Page 4, lines 71-2: Authors should provide full forms of abbreviations and shortforms when they first appear in the manuscript's main text, for example. *P. lilacinum* strain PLBJ-1 should be written as *Purpureocillium lilacinum* strain PLBJ-1.

Response: Thanks for your suggestion. It has been corrected. Please refer to the page 4, line 71.

16) Page 4, lines 73-5: Change ...structure elucidations of products demonstrated the LcsG conducts iterative methylation at the terminal-free amines of leucinostatins... to...structure elucidation of the products demonstrated that LcsG is involved in the iterative methylation of terminal-free amines in leucinostatins...

Response: Modifications have been made in accordance with your suggestion: “structural elucidation of the products demonstrated that LcsG is involved in the iterative methylation of terminal-free amines in leucinostatins”. Please refer to page 4, lines 74-75 for verification.

17) Page 8, line 150: Change ...with those data of LeuA (3)... to ...with those of LeuA (3)...

Response: Modifications have been made in accordance with your suggestion. Please refer to page 8, line 152 for verification.

18) Page 9, line 171: The authors should specify the purification technique so that readers can understand without having to refer to the methods section.

Response: Modifications have been made in accordance with your suggestion: “We purified these compounds by semipreparative HPLC and successfully obtained LeuA0 (**8**), LeuK0 (**4**) and LeuK3 (**7**) in sufficient quantities for the antimicrobial assay” Please refer to page 9, lines 173-175 for verification.

19) Page 9, line 185: The author state here: ...employed artificial intelligence (AI) methods to approximate a model of LcsG... The authors should specify which online AI predictive tool they used, such as Uni-Fold, and include a link to it in the methods section.

Response: Modifications have been made in accordance with your suggestion: “As an alternative, we employed artificial intelligence (AI) methods, Uni-Fold, to approximate a model of LcsG.” “Uni-Fold (<https://github.com/dptech-corp/Uni-Fold>) was employed to predict LcsG dimer structure.” Please refer to page 9, line 188-190 and page 23, line 497 for verification.

20) Page 10, line 194: When acronyms like SAH appear in the manuscript text for the first time, define them in their entirety.

Response: SAH appear in the manuscript text for the first time with its entirety in Page 6 line 119.

21) Page 10, lines 198 & 205: When using amino acid residue abbreviations in model structures, the authors should be consistent. In the text, Figures 4A–B, and S18, they seem to use single- and triple-letter abbreviations interchangeably.

Response: Modifications have been made in accordance with your suggestion. Please refer to page 10, lines 202-203 & 209 for verification. The corrected figure as below:

Figure 4:

Figure S18:

22) Page 10, line 209: Change ...showed decreases in the conversion of LeuA (3)... to ...showed a decrease in the conversion of LeuA (3)...

Response: Incorporating feedback from your comment and the editing service, this part has been changed to “the conversion of LeuA (3) to LeuA0 (8) decreased for all the mutants”. Please refer to pagex, line x for verification.

23) Page 10, lines 210-11: Change ...diffdock and MD simulations results... to ... docking and MD simulations results...

Response: Modifications have been made in accordance with your suggestion. Please refer to page 11, line 219 for verification.

24) Page 11, line 230: Change ...the LcsG-D368A and LcsGD395A showed obvious decreases... to ...the mutants D368A and D395A showed obvious decreased methylation...

Response: Modifications have been made in accordance with your suggestion. Please refer to page 12, line 237 for verification.

25) Page 12, line 241: Change ...capacity of catalyzing catalyzes two methylations... to ...capacity for catalyzing two methylations...

Response: Incorporating feedback from your comment and the editing service, this part has been changed to “catalyze the two methylation reactions”. Please refer to page 12, line 250 for verification.

26) Page 12, line 277: Change ...followed by released from... to...followed by release from...

Response: Modifications have been made in accordance with your suggestion. Please refer to page 14, line 286 for verification.

27) Page 15, line 306: Change ...filtration cultivation for... to ...filtration after cultivation for...

Response: Modifications have been made in accordance with your suggestion. Please refer to page 15, line 317 for verification.

28) Page 16, line 345: Change ...with 400 µg/mL G418 (Inalco, SA) (400 µg/mL) for 3-5 days... to ...with 400 µg/mL G418 (Inalco, USA) for 3-5 days...

Response: Modifications have been made in accordance with your suggestion. Please refer to page 17, line 356 for verification.

29) Page 18, line 391: Change ...dimethylformamide (THF)... to...dimethylformamide (DMF)...

Response: Modifications have been made in accordance with your suggestion. Please refer to page 19, line 403 for verification.

30) Page 23, line 485: Change Uni-fold to Uni-Fold and include the URL.

Response: Modifications have been made in accordance with your suggestion. Please refer to page 23, line 498 for verification.

31) Page 23, line 489: Change mafft and esript3 to MAFFT and ESPript 3 respectively, and include their URLs.

Response: Modifications have been made in accordance with your suggestion. Please refer to page 24, line 503-504 for verification.

32) Page 23, lines 494-98: Consider rephrasing to: ...The predicted LcsG dimer structure, and leucinostatin A structure was uploaded to initiate the docking process, and the website's backend algorithms processed the data to predict the ligand's binding poses to the protein. The model was then chosen from the final docking poses as well as the top ranked docking pose...

Response: This section has been changed to “The predicted LcsG dimer structure, and leucinostatin A structure were uploaded to initiate the docking process, and the website's backend algorithms were used to process the data and predict the ligand's binding poses to the

protein. The model was then chosen from the final docking poses as well as the top-ranked docking pose”. Please refer to page 24, line 509-513 for verification.

33) Page 24, line 510: Change ...E-Value threshold... to ...Evaluate threshold...

Response: Thank you for your suggestion. However, we have redone the SSN analysis, and as a result, the issue you mentioned no longer exists in the current version of the manuscript. We appreciate your attention to detail and guidance.

Reviewer #2 (Remarks to the Author):

1) I agree with the authors that the addition of sequence similarity analysis enhanced the manuscript. However, in the current form, evidence the support their hypothesis that NMTs are disguised as OMTs by simple sequence analysis is somewhat limited. The authors' arguments could be more compelling given some quick additional analysis. Firstly, the authors may be able to distinguish between OMTs and NMTs more easily by performing a new SSN with JUST the sequences from the orange cluster. Additionally, the authors discuss at length key differences in active site residues that distinguish OMTs from NMTs. It would be much more convincing if the authors manually curated their analysis to characterize sub-clusters within the orange cluster by these residues. The inclusion of sequence logos may also aid in the visualization of this analysis.

Response: Following your advice, we performed a SSN analysis focusing solely on the sequences within the orange cluster. The new Sequence Similarity Network (SSN) was generated using a group of experimentally verified methyltransferases, including 39 O-methyltransferases (OMTs) and 5 N-methyltransferases (NMTs), all of which belong to the same protein family (PF00891) as LcsG. (Please refer to Figure S27). The analysis of the SSN results reveals that NMTs do not form distinct independent clusters and exhibit connections with many OMTs. NMT Fsa4 clustered with three other NMTs (PynC, EqxD, and Phm5), likely due to similarities in their substrates (Fig. S27B).

Fig. S27:

We have also incorporated sequence logos for LcsG and the 39 OMTs to visually represent the amino acid residues at position Y367 and corresponding positions. This approach has indeed provided a more intuitive display of the differences between OMT and NMT residues at this critical site, compared to the original multiple sequence alignment (Please refer to Fig. S25).

Fig. S25

2) Do the authors think this is a case of convergent evolution? This question may be answered by a phylogenetic analysis.

Response: In response to whether the observed similarities between NMTs enzymes represent a case of convergent evolution, we did a phylogenetic analysis using LcsG and other reported methyltransferases. The result (Fig. S26A) indicates that OMT and NMT do not form distinct evolutionary branches but instead show closer genetic relationships within the same species (Fig. S26B). This pattern is indicative of speciation followed by gene differentiation within the species and the appearance of NMT could be the result of convergent evolution.

Fig. S26

A

B

	1	10	20	30	40	50	60
P16559	MAARTDNSIVVNAPFELVWDVTDNDIEAWPELSEYAEAEILRQDGDGDFRLKTRPDANG						
RosaA						
	70	80	90	100	110	120	
P16559	RVWEWVSHRVPDKGSRTVRAHRVETGPFAYMNLHWTYRAVAGGTEMRWVQEFDMKPGAPF						
RosaA						
	130	140	150	160	170	180	
P16559	DNAHMTAHLNNTTTRANMERIKKIIEDRHREGQTPASVLTTELHQLLLAASGRIRIRI						
RosaAMRPEPTHEPERTAAQRLYQYNVDLKVAFV						
	190	200	210	220	230	240	
P16559	VHVLTELRIA DLLADGPRHV AELAKE TD THELS LYRVLRSAASVGVFAEGEVRTE SATPL						
RosaA	LYAVAKLHL DLLADGPRTTADLAATGSDPSRLRLLRAAAGADALREVEEDSEELAFM						
	250	260	270	280	290		
P16559	SDGLRTGNPDGVLPLVKYNNMELTRRPYDEIMHSVRTGEPAFRRVFGSSFFELLEAN..P						
RosaA	GDLRSRSHPRSMRGMTTFFAEPDVLAA YGDLVESVRTGVPAFLRHREPLYDFLARPOHK						
	300	310	320	330	340	350	
P16559	EAGEFFERFMAHWSRRLVLDGLADQGMERFSRIADLGGDGWFLAQILRRRHPHATGILMD						
RosaA	EVRDEFDAAMVEFGQYFADDFLTSFDGFRFRFADLGGRCGFLAGVLTAVESSTGVLD						
	360	370	380	390	400	410	
P16559	LPRVAASAGPVLEEAKVADRVTVLP GDFFTDPVPTGYDAYLFRGVLEHNWSDERAVTVIRR						
RosaA	GPAAVAASAHKF LASQNLTERVEVRIGDEF.DVLP TGC DAYVLRGVLEDWADADAVRLLVR						
	420	430	440	450	460	470	
P16559	VREAIIGDD.DARLLIFDOVMAPEN EWDHAKLLDIDMLVLFGGRRVLA EWRQLLEADFD						
RosaA	IRQAMGDAP EARLLI LDSVI GETGE..L GKVLIDMLV LVEGERRTRACWDDLLARAGFD						
	480	490					
P16559	IVNT..PSHTWTTLECRPV..						
RosaA	IVGIHPAGDVWAVLECRGTAG						

Additional minor questions/critiques are detailed below.

- 1) The authors provided an explanation of their antifungal assays along with why certain strains were selected in their point-by-point response but did not alter the text itself. Without any clarification, this remains confusing for the reader. By no means am I suggesting that the authors incorporate their lengthy response, but minor changes could help clarify this section. E.g. "Leucinostatins are well-known [antimicrobial and antimycotic compounds]. We carried out the purification for the compounds and obtained LeuA0 (8), LeuK0 (4) and LeuK3 (7) in sufficient quantities to do the antifungal assay. We determined the inhibitory activity of these compounds against the drug resistant strain *C. neoformans* H99 and the plant pathogen *P. infestans*, [both of which have been shown to be inhibited by LeuA and LeuB,(references)] by agar diffusion assays."

Response: Incorporating feedback from your comment and the editing service, this part has been changed to "Leucinostatins are well-known antibiotics. We purified these compounds by semipreparative HPLC and successfully generated LeuA0 (8), LeuK0 (4) and LeuK3 (7) in sufficient quantities for the antimicrobial assay. The inhibitory activity of these compounds against the drug-resistant strain *C. neoformans* H99 and the plant pathogen *P. infestans* was determined using agar diffusion assays. Notably, previous studies have demonstrated the inhibitory effects of LeuA and LeuB on both *C. neoformans*¹⁸ and *P. infestans*²⁶". Please refer to page 9, lines 1173-178 for verification.

- 2) As DiffDock is a very niche software, the term 'molecular docking' seems more appropriate for use in the abstract. E.g. "Molecular docking and dynamics simulations showed...". Likewise, the second 'D' in 'DiffDock' is also capitalized.

Response: Modifications have been made in accordance with your suggestion. Please refer to page 2, line 28 for verification.

- 3) The text still does not read wonderfully smoothly and there are a variety of grammatical and typographical errors. Perhaps it would be worth investigating an editing service. Two examples that are immediately apparent in the introduction are below, but there are others throughout.

"One study that discusses the N-methylation of the pipercolic acid moiety, which serves as the initial substrate for tubulysin synthesis." This is not a complete sentence, so it is unclear what the authors are trying to convey.

"To our knowledge, a discrete NMT-catalyzed iterative Nmethylation is not previously observed in the terminus of NRPs." The verb tenses are mixed here, along with a missing pluralization. Something like "To our knowledge, a discrete NMT-catalyzed iterative N-methylation has not previously been observed in the termini of NRPs." Would be clearer.

res

Response: "One study that discusses the N-methylation of the pipercolic acid moiety, which serves as the initial substrate for tubulysin synthesis." has been changed to "One study explores

the *N*-methylation of the pipercolic acid moiety, which serves as the initial substrate for tubulysin synthesis.” Please refer to page 3, lines 52-53 for verification.

Incorporating feedback from your comment and the editing service, “To our knowledge, a discrete NMT-catalyzed iterative Nmethylation is not previously observed in the terminus of NRPs.” has been changed to “To our knowledge, the termini of NRPs have never before been shown to undergo a discrete NMT-catalyzed iterative N-methylation.” Please refer to page 3 &4, lines 54-56 for verification.

4) The links in the SI table of contents are not all associated with the correct sections in the document.

Response: We have checked and confirmed the page numbers associated with the links in the SI table of contents to ensure that they correspond correctly to the respective sections in the document.”

5) The additional figures depicting the complete structure of the enzyme are certainly worthwhile, but I nonetheless encourage the authors to label the termini as the roles of both are described in the text.

Response: The figure has been incorporated into the SI as Figure S18B. Please see it.

Figure S18B:

6) Upon discussing the pLDDDT scores of the overall structure as well as specific residues on page 9, the authors could make their argument more easily digestible by clearly stating that AI predicts the structure with high confidence, including active site residues proposed to be relevant for substrate binding and catalysis.

Response: Modifications have been made in accordance with your suggestion, “These results indicated that AI predicts the protein structure with high confidence, including the active site

residues that are proposed to be relevant for substrate binding and catalysis”. Please refer to page 10, lines 192-194 for verification.

- 7) Did the authors mean ‘subsequent molecular dynamics’ when they said “The following molecular dynamics (MD) simulations”? ‘Following’ implies they will talk about those simulations immediately after that sentence, but that is not the case.

Response: Thank you for your comment regarding the phrasing "the following molecular dynamics (MD) simulations" to “subsequent molecular dynamics (MD) simulations”. Modifications have been made in accordance with your suggestion. Please refer to page 10, line 209 for verification.

- 8) On line 223, the authors use mutant data as support for their argument, but this is before the mutant data have been discussed (primarily lines 230-231). If the authors simply remove reference to the mutant data in line 223, the paragraph will likely read more smoothly. The introduction of that data in lines 230-231 follows nicely from the computational discussion.

Response: Modifications have been made in accordance with your suggestion. Please refer to page 12, lines 238-240 for verification.

- 9) The authors describe ‘boxes’ in their SSN. Is this outdated? Their current SSN seems to be organized by color, not boxes.

Response: Thank you for your comment highlighting the confusion. My original intention was to describe the shape of each node in the SSN as square “boxes”. However, based on your feedback, we have revised the SSN diagram to avoid this issue altogether.

REVIEWERS' COMMENTS:

Reviewer #1 (Remarks to the Author):

The manuscript is much clearer now than it was before. The current version has been greatly improved, with corrections and the addition of missing information. However, one minor correction is required before it can be considered for publication.

Page 9, lines 178-79: Change ... All of these leucinostatins inhibited effects against these pathogens (Fig. 3D)... to ...All of these leucinostatins had inhibitory effects on these pathogens (Fig. 3D)...

Reviewer #2 (Remarks to the Author):

The clarity and readability of the manuscript are much improved, and my previous suggestions/concerns were suitably addressed. Additional clarifying sentences to sum up the conclusions associated with the new phylogenetic and sequence analysis would still be beneficial. For example:

- Lines 282-284: "Based on sequence similarity, we propose that in addition to LcsG, those labeled as OMTs available on the genomic databases that rely on sequence similarity, which might function as NMTs." This sentence doesn't make sense grammatically. Are the authors trying to state that sequences annotated as OMTs may actually be NMTs and should be evaluated for the presence of Y versus H in the key position to assess this? If so, clarification of this sentence and perhaps the addition of another would help emphasize the impact of this bioinformatic analysis. Likewise, a sentence discussing/summarizing their conclusions regarding the evolution of NMTs and OMTs would also be helpful.

Otherwise, some typos/grammatical errors/confusion remain. I encourage the authors to carefully read through the text and captions one more time to address some of these issues. See some examples I noticed below, although others likely exist.

Main

- Abstract "Molecular docking and the molecular dynamics simulations showed that..." The article before "molecular dynamics is unnecessary.
- Abstract "highly-conserved" does not need the hyphen
- Line 51 "singlesite" is not typically written as one word. "single-site" might be more appropriate.
- Line 196 "ClassI" should be "Class I"
- Lines 274-275 "this result indicated that a gene duplication event may lead to divergence of OMT and NMT after the speciation of Streptomyces." Are the authors trying to talk about what happened in the past? If so then "...event may have lead to..." would be more appropriate.

SI

- Table S5 caption "methyltransfers" should be two words.
- Organisms in Table S7 should be italicized.
- Figure S18 caption "dicipeted" should read "depicted"

Characterization of a methyltransferase for iterative *N*-methylation at the leucinostatin termini in *Purpureocillium lilacinum*

Reviewer #1:

1) Page 9, lines 178-79: Change ... All of these leucinostatins inhibited effects against these pathogens (Fig. 3D)... to ...All of these leucinostatins had inhibitory effects on these pathogens (Fig. 3D)...

Response: Modifications have been made in accordance with your suggestion. Please refer to page 9, lines 178-179.

Reviewer #2:

1) Lines 282-284: “Based on sequence similarity, we propose that in addition to LcsG, those labeled as OMTs available on the genomic databases that rely on sequence similarity, which might function as NMTs.” This sentence doesn’t make sense grammatically. Are the authors trying to state that sequences annotated as OMTs may actually be NMTs and should be evaluated for the presence of Y versus H in the key position to assess this? If so, clarification of this sentence and perhaps the addition of another would help emphasize the impact of this bioinformatic analysis. Likewise, a sentence discussing/summarizing their conclusions regarding the evolution of NMTs and OMTs would also be helpful.

Response: Following your advice, the sentence has been revised to: “Apart from LcsG, we propose that some sequences annotated as OMTs in genomic databases may actually be NMTs. The difference of some specific basic residues at key positions in OMTs and NMTs, such as the Y versus H, can be served as an indicator.”

Additionally, we have included the following summarizing sentence: “This bioinformatic analysis indicated a possible evolutionary link between OMTs and NMTs, and also emphasized the important functions of those specific amino acids in OMTs and NMTs.

Please refer to page 14, lines 288-293 for verification.

2) Otherwise, some typos/grammatical errors/confusion remain. I encourage the authors to carefully read through the text and captions one more time to address some of these issues. See some examples I noticed below, although others likely exist.

Main:

- Abstract “Molecular docking and the molecular dynamics simulations showed that...” The article before “molecular dynamics is unnecessary.
- Abstract “highly-conserved” does not need the hyphen
- Line 51 “singlesite” is not typically written as one word. “single-site” might be more appropriate.
- Line 196 “ClassI” should be “Class I”

- Lines 274-275 “this result indicated that a gene duplication event may lead to divergence of OMT and NMT after the speciation of Streptomyces.” Are the authors trying to talk about what happened in the past? If so then “...event may have lead to...” would be more appropriate.

SI:

- Table S5 caption “methyltransfers” should be two words.
- Organisms in Table S7 should be italicized.
- Figure S18 caption “dicipeted” should read “depicted”

Response: We have made the corrections you pointed out. Please refer to the following locations in the main text and supplementary information for the changes:

Main Text:

Page 2, line 27

Page 2, line 26

Page 3, line 49

Page 10, line 197

Page 13, line 281

Supplementary Information:

Supplementary Table 5

Supplementary Table 7

Supplementary Figure 18